# Convex Optimization for Alignment and Preference Learning on a Single GPU

**Miria Feng** [1]  **Mert Pilanci** [1]

## Abstract

Fine-tuning large language models (LLMs) to align with human preferences has driven the success of systems such as Gemini and ChatGPT. However, approaches like Reinforcement Learning from Human Feedback (RLHF) remain computationally expensive and complex. Direct Preference Optimization (DPO) offers a simpler alternative but has limitations such as inconsistent ranking accuracy, high dependence on GPU resources, and expensive hyperparameter tuning. We propose the **C**onvex **O**ptimization for **A**lignment and Preference **L**earning **A**lgorithm (**COALA**): a novel lightweight strategy with strong theoretical guarantees. By leveraging the convex optimization reformulation of neural networks, COALA eliminates the need for a reference model and obtains significant reduction in both training time and VRAM consumption, thus enabling efficient training on a single GPU. Experiments across four datasets—including a 26 621-sample synthetic Educational Feedback dataset—and six models (including Llama-3.1-8B) demonstrate COALA's competitive performance and efficiency while utilizing as little as $\sim17.6\%$ of DPO's total TFLOPs. COALA exhibits stable, monotonically increasing rewards and reaches peak margins in significantly shorter time in comparison to traditional methods such as DPO and ORPO. To the best of our knowledge, this is the first time convex optimization has been effectively applied to preference fine-tuning of LLMs.

## 1. Introduction

Large Language Models (LLMs) have been trained on increasing amounts of data to capture semantic language pat-

---

[1]Department of Electrical Engineering, Stanford University, California, United States. Correspondence to: Miria Feng <miria00@stanford.edu>.

*Proceedings of the 43rd International Conference on Machine Learning*, Seoul, South Korea. PMLR 306, 2026. Copyright 2026 by the author(s).

terns and scale to solve more complex problems. The main paradigm combines pre-training and fine-tuning LLMs to achieve aligned user-preferable responses, and train personalized Language Model (LM) assistants for widespread applications (Google DeepMind, 2023; OpenAI, 2022; Anthropic, 2023). Reinforcement Learning from Human Feedback (RLHF) (Stiennon et al., 2020; Ouyang et al., 2022; Christiano et al., 2017; Wang et al., 2023) demonstrates impressive results for preference alignment, and utilizes a three-step approach of Supervised Fine-Tuning (SFT), reward model training, and policy optimization. However, its complex multi-stage methodology presents several optimization challenges, requires humans in the loop, and is notably compute resource-intensive. This has driven timely and significant research on resource-efficient preference learning strategies, especially as energy demands outpace infrastructure (Chen et al., 2025).

The Direct Preference Optimization (DPO) (Rafailov et al., 2024) algorithm proposes a simpler and more computationally lightweight alternative to aligning LLMs for user-preferred responses. DPO parametrizes the reward function instead of learning an explicit reward model and incorporates this into the Bradley-Terry ranking objective (Bradley & Terry, 1952). Although simpler, this also yields certain drawbacks: requiring a reference model to stabilize training incurs additional memory costs to host two models in order to train a single model, the approximately $10\times$ smaller learning rate (versus SFT) (HuggingFace Alignment Team, 2023) slows optimization, while a mismatch between the reward optimized in training and the log-likelihood optimized during inference can lead to unstable reward margin gains (Meng et al., 2024).

To address these issues, recent studies have proposed eliminating the reference model to reduce memory overhead (Jung et al., 2025; Meng et al., 2024), advanced deeper theoretical understanding on the behavior of RLHF and DPO with in-depth analysis (Azar et al., 2024; Ethayarajh et al., 2024; Swamy et al., 2025), and reduced dependence on the prerequisite SFT stage, which is typically imperative to convergence (Hong et al., 2024; Xu et al., 2024). However, these techniques also introduce additional brittle hyperparameters, which the authors note are *crucial* to achieving good performance, or require small learning rates on the order of $1\times10^{-9}$ for stable convergence. The large reliance on

heuristic-driven methods may also result in random model ranking accuracy even after time-extensive training (Chen et al., 2024).

We present COALA, a fast and lightweight framework for effective preference alignment of LLMs on a single GPU. COALA leverages a *convex* two-layer neural network (cvxNN) on top of a pre-trained model for fast convergence with theoretical guarantees, and thus mitigates the instability of offline alignment via consistent reward margin increments. Unlike existing DPO style objectives, COALA utilizes an Alternating Direction Method of Multipliers (ADMM) (Boyd et al., 2011) approach to enable scalable, near-hyperparameter-free optimization. We implement COALA in JAX (Bradbury et al., 2018) and efficiently execute Just-In-Time (JIT) compilation to maximize VRAM throughput. As a result, COALA fine-tunes models such as Llama-3.1-8B on a single RTX-4090 GPU (NVIDIA Corporation, 2025) and demonstrates competitive performance on multiple benchmarks, including AlpacaEval2 (Li et al., 2023), ArenaHard (Li et al., 2024), and MT-Bench (Zheng et al., 2023). Comprehensive experiments span six models × four datasets × four methods. Results presented total over **216 training runs**, and TFLOPs metrics further validate COALA's substantial compute and power efficiency. Since auto-run benchmarks are ultimately surrogate for measuring real human preference, we conduct fair double-blind evaluation with **107 real human samples** to assess the validity of benchmark scores against actual human feedback, and establish COALA's practical effectiveness in a real world scenario. Our main contributions are summarized as follows:

- We introduce COALA: a more interpretable and theory-backed convex framework that leverages principled optimization to enable effective, single-GPU preference alignment.

- We prove COALA's convergence guarantees. This theoretical analysis supports COALA's exhibited stable convergence, while mitigating reliance on hyperparameter tuning and broad heuristics.

- We present EduFeedback[1]: the open-source dataset comprised of 26 621 student-tutor conversations. We additionally introduce the **"Alternating Population Strategy"**, to uniquely derive 65 606 preference training pairs, eliminating the need for external re-ranking models and establishing a more efficient paradigm for preference dataset synthesis.

- Our modular open-source JAX codebase[2] ensures ease

---

[1] https://huggingface.co/datasets/miria0/EduFeedback
[2] https://github.com/pilancilab/COALA

of reproducibility for ongoing research, as well as practical on-premises single GPU use cases.

## 2. Related Work

Preference fine-tuning LLMs can be classified as three distinct strategies. **(1)** Initial algorithms of zero-shot and few-shot in-context learning (Brown et al., 2020) rely on prompt engineering. This method does not require extensive compute, but is unable to tackle complex tasks due to reliability issues. **(2)** More sophisticated algorithms use reinforcement learning to align model output with user preferences. Successful algorithms in this class (such as RLHF and RLAIF (Lee et al., 2023)) have been able to create conversational LLMs such as Google Gemini (Google DeepMind, 2023) and ChatGPT (OpenAI, 2022). However, despite their impressive performance, the high computational costs of these methods pose a substantial barrier to entry for many applications. **(3)** DPO-style methods leverage the Bradley & Terry (1952) model to provide simple yet often performant learning algorithms that do not require explicit reward modeling. Yet good results based on these methods consistently rely on extensive hyperparameter grid-search (Meng et al., 2024), and are largely heuristics-driven.

As LLMs become more widely adopted, there is also escalating awareness that scaling compute alone is increasingly unsustainable economically, environmentally, and in terms of diminishing returns (Singh et al., 2025). This has motivated recent work to make progress through avenues such as repeated sampling (Brown et al., 2024), creative chain-of-thought reasoning (Wei et al., 2022), and theoretically interpretable techniques (Singh et al., 2023). Single GPU methods are a timely and practical direction for preference fine-tuning, especially as local accelerators become increasingly powerful (Apple Inc., 2025). Targeting a single device improves robustness (fewer distributed failure modes), latency (no cross-node synchronization), and data locality/security (privacy-sensitive deployment with on-premises options). This trend is reflected in recent systems that deliver machine learning applications in text, vision, and multimodal settings (Sheng et al., 2023; Geiping & Goldstein, 2023; Ning et al., 2024; Kwon et al., 2020; Tragakis et al., 2024). Human preference alignment is ideally positioned in this emerging paradigm, which seeks low-latency, resource-aware algorithms without sacrificing generative quality.

Bengio et al. (2005) have previously shown that it is possible to characterize the optimization problem for neural networks as a convex program. Pilanci & Ergen (2020) further developed exact convex reformulations for training a two-layer ReLU neural network. The reformulation uses semi-infinite duality theory to show that two-layer neural networks (NNs) with ReLU activations and weight decay regularization may be expressed as a linear model with group lasso penalty

and polyhedral cone constraints. This yields both practical benefits in implementation and theoretical advantages in analyzing the optimization of non-convex landscapes in NNs. Bai et al. (2023) and Feng et al. (2024) have recently proposed cvxNN methods based on ADMM techniques to solve high-dimensional deep learning tasks. ADMM offers several advantages, such as its robustness against hyperparameter selection, linear decomposability for distributed optimization, and immunity to vanishing/exploding gradients. Its natural parallelization ability makes ADMM particularly suitable for optimization in deep learning, where scalability is crucial. The COALA algorithm combines these features to effectively preference fine-tune LLMs faster with less memory and compute while demonstrating competitive performance.

# 3. Convex Neural Networks

This section provides background on convex neural networks (cvxNN), which is the basis for our introduction of the COALA algorithm. Our approach is motivated by defining a principled preference fine-tuning objective with interpretable convergence guarantees.

## 3.1. Two-layer ReLU Networks

Given input $x \in \mathbb{R}^d$, the classic two-layer ReLU network is given by:

$$f(x) = \sum_{j=1}^{m} (\Theta_{1j} x)_+ \theta_{2j}, \qquad (1)$$

where $\Theta_1 \in \mathbb{R}^{m \times d}$, $\theta_2 \in \mathbb{R}^m$ are weights of the first and last layer respectively, and $(\cdot)_+ = \max\{\cdot, 0\}$ is the ReLU activation function.

Given targets $y \in \mathbb{R}^n$, the network in (1) is trained by minimizing the following non-convex loss function:

$$\min_{\Theta_1, \theta_2} \ell\left(f_{\Theta_1, \theta_2}(X), y\right) + \frac{\beta}{2} \sum_{j=1}^{m} \left(\|\Theta_{1j}\|_2^2 + (\theta_{2j})^2\right), \quad (2)$$

where $\ell : \mathbb{R}^n \mapsto \mathbb{R}$ is the loss function, $X \in \mathbb{R}^{n \times d}$ is the data matrix, and $\beta \geq 0$ is the regularization strength. The non-convex nature of (2) makes its solution challenging, since the optimizer typically needs meticulous tuning of hyperparameters to ensure successful training. This requires many expensive iterations of running the optimizer across multiple hyperparameter configurations in a grid search to obtain good performance. This dramatically contrasts with the convex optimization framework, where algorithms have strong convergence guarantees and involve minimal hyperparameters. It is desirable to maintain the expressive capabilities of such neural networks while still preserving the computational advantages of convex optimization.

## 3.2. Convex reformulation

Pilanci & Ergen (2020) have shown (2) admits a convex reformulation, thus alleviating the inherent difficulties of the non-convex landscape in deep learning. When condition $m \geq m^*$ is satisfied for some $m \geq n + 1$, the reformulation has the same optimal value as the original non-convex problem and no information is lost.

The convex reformulation is based on enumerating the actions of all possible ReLU activation patterns on the data matrix $X$. These activation patterns act as separating hyperplanes, which essentially multiply the rows of $X$ by 0 or 1 and can be represented by diagonal matrices. For fixed $X$, the set of all possible ReLU activation patterns may be expressed as

$$\mathcal{D}_X = \left\{ D = \mathrm{diag}\left(\mathbf{1}(Xv \geq 0)\right) : v \in \mathbb{R}^d \right\}.$$

The cardinality of $\mathcal{D}_X$ grows as $|\mathcal{D}_X| = \mathcal{O}\left(r(n/r)^r\right)$, where $r := \mathrm{rank}(X)$ (Pilanci & Ergen, 2020). Given $D_i \in \mathcal{D}_X$, the set of vectors $v$ for which $(Xv)_+ = D_i Xv$, is given by the following convex cone: $\mathcal{K}_i = \{v \in \mathbb{R}^d : (2D_i - I)Xv \geq 0\}$.

The exponential size of $\mathcal{D}_X$ make its complete enumeration impractical. Instead, we work with a subset based on sampling $P$ patterns from $\mathcal{D}_X$ for tractability:

$$\min_{(v_i, w_i)_{i=1}^P} \ell\left(\sum_{i=1}^{P} D_i X(v_i - w_i), y\right) + \beta \sum_{i=1}^{P} \|v_i\|_2 + \|w_i\|_2$$
$$\text{s.t. } v_i, \ w_i \in \mathcal{K}_i \quad \forall i \in [P].$$
$$(3)$$

Although (3) is based on subsampling patterns in the convex reformulation, it can be shown under mild conditions that (3) still has the same optimal solution as (2) (Mishkin et al., 2022). The recent work of Kim & Pilanci (2024) also prove that the difference is negligible even when they are not equal. Therefore we can confidently work with the convex program in (3).

In this paper we denote $\ell$ to be the mean-square error loss. Recent work (Bai et al., 2018) has shown that by adding slack variables, (3) can be written as:

$$\min_{v,s,u} \|Fu - y\|_2^2 + \beta \|v\|_{2,1} + \mathbb{I}_{\geq 0}(s) \quad \text{s.t. } u = v, \ Gu = s$$
$$(4)$$

The matrix $F \in \mathbb{R}^{n \times 2dP}$ is block-wise constructed by $D_i X$ terms.

# 4. COALA

In this section, we introduce comprehensive methodology of the **C**onvex **O**ptimization for **A**lignment and Preference **L**earning **A**lgorithm (**COALA**). Appendix B gives a more explicit step-by-step through the COALA method.

## 4.1. Convex Preference Optimization Framework

In standard DPO, the goal is to obtain a good policy that is aligned with human preferences. The policy network $\pi_\theta$ is first initialized with the weights of a pre-trained network. It then aligns the policy model by solving the optimization problem

$$L_{\text{DPO}}(\pi_\theta; \pi_{\text{ref}}) = \tag{5}$$
$$- \mathbb{E}\left[\log \sigma \left(\beta \log \frac{\pi_\theta(y_w|x)}{\pi_{\text{ref}}(y_w|x)} - \beta \log \frac{\pi_\theta(y_l|x)}{\pi_{\text{ref}}(y_l|x)}\right)\right].$$

The DPO objective in (5) is a large-scale non-convex optimization, which is challenging to solve. We first reformulate this learning problem using cvxNN, in order to admit more elegant optimization techniques. We adopt a modified Bradley-Terry model with an offset parameter $\gamma > 0$.

$$p(y_w \succ y_l | x) = \sigma(r(x, y_w) - r(x, y_l) - \gamma), \tag{6}$$

COALA then uses the un-normalized log-likelihood as its rewards function.

$$r(x, y) = \beta \log \pi(y|x). \tag{7}$$

Instead of taking $\pi$ in (7) to be a traditional neural network (NN) model, we replace it with a cvxNN. Specifically, we take a pre-trained model $f_{\theta_{\text{pre}}}(x)$ and stack a two-layer convex neural network $g^{\text{cvx}}_{\Theta_1, \theta_2}$ on top to serve as a binary preference classifier. $g^{\text{cvx}}_{\Theta_1, \theta_2}$ is then trained by solving the convex optimization problem in (4). Letting $\theta = (\theta_{\text{pre}}, \Theta_1, \theta_2)$, the resulting policy is then given by:

$$\pi^{\text{cvx}}_\theta(y|x) := \frac{1}{1 + \exp\left(-y g^{\text{cvx}}_{\Theta_1, \theta_2}\left(f_{\theta_{\text{pre}}}(x)\right)\right)}.$$

Instead of executing preference optimization with the weights of the entire model $g^{\text{cvx}}_{\Theta_1, \theta_2} \circ f_{\theta_{\text{pre}}}$, we freeze the weights of $f_{\theta_{\text{pre}}}$ and freeze $\Theta_1$ in $g^{\text{cvx}}_{\Theta_1, \theta_2}$. Inserting (7) into (6) with $\pi(y|x)$ replaced by $\pi^{\text{cvx}}_\theta(y|x)$, yields the COALA objective:

$$\min_{\theta_2} \; L_{\text{COALA}}(\pi^{\text{cvx}}_{\theta_2}) := \tag{8}$$
$$- \mathbb{E}_{(x, y_w, y_l) \sim \mathcal{D}} \left[\log \sigma \left(\beta \log \frac{\pi^{\text{cvx}}_{\theta_2}(y_w|x)}{\pi^{\text{cvx}}_{\theta_2}(y_l|x)} - \gamma\right)\right],$$

Notably, (8) is reference-free in comparison to (5) yet does not incur additional scaling hyperparameters for stability. This is supported in recent work (Hong et al., 2024; Meng et al., 2024; Gupta et al., 2024), where we find the reference model unnecessary.

In addition to improved stability and memory efficiency by removing dependence on the reference model, the advantage of solving (8) over (5) also gives higher computational tractability. The following proposition shows (8) is convex.

**Proposition 4.1** (COALA Loss is Convex). *The optimization problem in* (8) *may be written as*

$$\min_{\theta_2} \; \mathbb{E}\left[\log\left(1 + \exp\left(-\beta y_w \theta_2^T (\Theta_1 f_{\theta_{\text{pre}}}(x))_+ + \gamma\right)\right)\right]. \tag{9}$$

*This objective is convex as* (9) *is a logistic regression problem in* $\theta_2$.

The proof of Theorem 4.1 is provided in Section A. Theorem 4.1 shows that $L_{\text{COALA}}$ is convex. Thus, we can solve it to global optimality in polynomial time using more efficient gradient-based optimizers. Since we only have to train the weights of the final layer of the convex model, COALA requires significantly less computation than existing methods and provides the second major reason why COALA can quickly train on one GPU.

In summary, throughout our theoretical development the probability object induced by COALA is the *pairwise preference probability* used in the Bradley–Terry model, rather than the full autoregressive language-model sequence probability. More precisely, COALA models the Bernoulli event that a preferred response $y_w$ is ranked above a rejected response $y_l$ conditioned on the prompt $x$, and the corresponding logistic form is derived only in this pairwise-comparison setting. The notation $\pi^{\text{cvx}}_\theta(y \mid x)$ is therefore shorthand for a *preference score/probability* over a comparison outcome, not for the frozen base model's sequence distribution over complete text responses.

## 4.2. COALA Algorithm

We formally present the pseudocode for COALA in Algorithm 1. Stage one of COALA first trains a cvxNN on top of a standard pre-trained model. In stage two, the final policy model is obtained by solving a convex logistic regression problem in a fine-tuning step.

The key to maximizing COALA's efficiency in this framework is using the recently introduced CRONOS algorithm (Feng et al., 2024) to train $\pi^{\text{cvx}}_\theta$, which we now discuss in detail.

### 4.2.1. EFFICIENTLY TRAINING THE CONVEX POLICY NETWORK VIA CRONOS

CRONOS (Feng et al., 2024) is a specialized version of the Alternating Direction Method of Multipliers (ADMM) (Boyd et al., 2011), for solving the optimization problem in (4). The choice of ADMM for solving (4) in Feng et al. (2024) is predicated on three key properties: **(1)** It possesses a robust convergence guarantee as shown in Section 4.3, **(2)** it lifts memory constraints in LLM classification problems, and **(3)** it is highly optimized for GPU acceleration and parallelism in JAX.

Algorithm 2 formally presents using CRONOS for solving

---

**Algorithm 1** Convex Preference Optimization (COALA)

---

**input** Dataset $(x, y_w, y_l)$, Pre-trained model $f_{\theta_{\text{pre}}}(x)$, offset parameter $\gamma$, penalty parameter $\rho > 0$

    **Phase I**: **Train the policy network**

    Train $\pi_\theta^{\text{cvx}}$ to obtain $(\Theta_1, \theta_2)$ by solving (4) using CRONOS$(\rho)$ (Algorithm 2).

    **Phase II**: **Finetuning**

    Freeze weights of the first layer $\Theta_1$.

    Finetune weights of second layer $\theta_2$ by solving the convex minimization problem (9):

$$\min_{\theta_2} \; L_{\text{COALA}}(\pi_{\theta_2}^{\text{cvx}}) \quad \{\text{Solve via AdamW}\}$$

**output** $(\Theta_1, \theta_2)$.

---

**Algorithm 2** CRONOS with Preconditioned Conjugate Gradient (PCG)

---

**input** penalty parameter $\rho$

    **repeat**

$$u^{k+1} = \arg\min_u \{\& \tfrac{1}{2}\|Fu - y\|^2 + \tfrac{\rho}{2}\|u - v^k + \lambda^k\|_2^2 + \tfrac{\rho}{2}\|Gu - s^k + \nu^k\|^2\} \; \{\text{Use PCG}\}$$

$$\begin{bmatrix} v^{k+1} \\ s^{k+1} \end{bmatrix} = \arg\min_{v,s} \{\beta\|v\|_{2,1} + \mathbf{1}(s \geq 0) + \tfrac{\rho}{2}\|u^{k+1} - v + \lambda^k\|^2\} \; \{\text{Primal update}\}$$

$$\lambda^{k+1} \leftarrow \lambda^k + \tfrac{\gamma_\alpha}{\rho}(u^{k+1} - v^{k+1}) \; \{\text{Dual } \lambda \text{ update}\}$$

$$\nu^{k+1} \leftarrow \nu^k + \tfrac{\gamma_\alpha}{\rho}(Gu^{k+1} - s^{k+1}) \; \{\text{Dual } \nu \text{ update}\}$$

    **until** convergence

---

Equation (4). Two subproblems must first be solved efficiently, then the algorithm only requires vector addition and one matrix-vector product. The structure of these subproblems makes CRONOS readily compatible with local hardware accelerators and JAX.

**What COALA optimizes.** COALA does not perform gradient-based updates to the pretrained language model weights. Instead, it freezes the base LLM and learns a convex neural network (cvxNN) head on top of frozen pretrained features. This design is deliberate: by restricting optimization to the convex head, we obtain global optimality guarantees for the trained preference module, substantially reduce memory and compute requirements, and preserve a practical single-GPU training regime. Consequently, COALA should be interpreted as a preference alignment framework built on top of frozen LLM representations, rather than as full-model policy optimization in the style of standard DPO fine-tuning.

### 4.3. COALA Convergence Guarantees

Since COALA interprets the preference alignment task as a convex optimization problem application, it also immediately inherits the rich convergence theory associated with convex algorithms. In particular, we show that this leads

to convergence guarantees for each aspect of the COALA pipeline and robustness to hyperparameter tuning. The following result (Feng et al., 2024) shows CRONOS converges ergodically at an $\mathcal{O}(1/k)$-rate.

**Theorem 4.2** (Convergence of ADMM for (4)). *Let $\{\delta_k\}_{k \geq 1}$ be some summable sequence. Run Algorithm 2 and suppose at each iteration the computed $u^{k+1}$ satisfies:*

$$\left\| u^{k+1} - \arg\min_u \left\{ \frac{1}{2}\|Fu - y\|^2 \right.\right.$$
$$+ \frac{\rho}{2}\|u - v^k + \lambda^k\|_2^2$$
$$\left.\left. + \frac{\rho}{2}\|Gu - s^k + \nu^k\|^2 \right\} \right\| \leq \delta_k.$$

*Then after $K$ iterations, the output of CRONOS Algorithm 2 satisfies:*

$$\|F\bar{u}^K - y\|^2 + \beta\|\bar{v}^K\|_{2,1} + \mathbf{1}(\bar{s}^K \geq 0) - p^\star = \mathcal{O}(1/K),$$
$$\left\| \begin{bmatrix} I_{2dP} \\ G \end{bmatrix} \bar{u}^K - \begin{bmatrix} \bar{v}^K \\ \bar{s}^K \end{bmatrix} \right\| = \mathcal{O}(1/K).$$

This guarantee holds for any $\rho > 0$ and when the $u$-subproblem is solved inexactly. Consequently, CRONOS' convergence is robust, making it an ideal subroutine for training the cvxNN in COALA. This ensures efficient and successful training of the policy network.

The COALA fine-tuning step also has strong guarantees, since the minimization objective in (9) is smooth, convex, and has a **Lipschitz continuous gradient**. The latter property follows as the logistic loss has a Lipschitz continuous gradient. Thus, if we apply Accelerated Gradient Descent (AGD) (Nesterov, 1983; d'Aspremont et al., 2021), which has the worst-case optimal convergence rate, to solve (9), we obtain the following result.

**Theorem 4.3** (Efficient minimization of COALA loss Equation (9)). *Suppose we run AGD to solve (9). Then after $k$ iterations, the output $\theta_2^k$ satisfies:*

$$L_{\text{COALA}}(\pi_{\theta_2^k}^{\text{cvx}}) - \min_{\theta_2} \; L_{\text{COALA}}(\pi_{\theta_2}^{\text{cvx}}) = \mathcal{O}(1/k^2).$$

Theorem 4.3 shows we can train the COALA loss to global optimality in polynomial time. This contrasts greatly with DPO variants, which are non-convex and lack convergence guarantees. In addition, optimizer methods in DPO-based approaches are typically non-trivial, sensitive to hyperparameter tuning, and typically require dramatically smaller learning rates than their respective SFT methods.

## 5. Experiments

We experiment with six models: DistilGPT-2 (Sanh et al., 2019), GPT-2 (Radford et al., 2019), Mistral-7B (Jiang et al.,

2023a), Dolphin-2.6-7B (Cognitive Computations, 2024), Llama-3.2-3B (Grattafiori et al., 2024), and Llama-3.1-8B (Grattafiori et al., 2024) × four datasets across four preference alignment algorithms. In order to ensure fair comparison, we run each individual experiment configuration on a single A100 GPU with 40GB VRAM, with the exception of COALA experiments which run on a single RTX-4090 GPU with 24GB VRAM. This ablation is necessary, since methods such as DPO and ORPO require more memory on the same datasets (Appendix E.1). We also perform controlled initialization study with SFT baselines (versus non-SFT init). All subsequent numerical results are presented after averaging over three runs in each setting, final evaluations are performed on held-out prompts. Tables 1 and 2 summarize main model and dataset features.

| Model | Parameters | Feature |
|---|---|---|
| DistilGPT-2 | 82M | Mini |
| GPT-2 | 124M | Seminal |
| Llama-3.2-3B | 3B | Modern |
| Mistral-7B | 7B | Performant and Balanced |
| Dolphin-2.6-7B | 7B | Instruction Fine-tuned |
| Llama-3.1-8B | 8B | Largest |

*Table 1.* Model sizes and characteristics evaluated in our study.

## 5.1. Datasets

This section briefly introduces the four main datasets of Table 2, with additional preference data extraction details presented in Appendix E.2.

**EduFeedback** is our synthetically generated conversational dataset inspired by UltraChat (Ding et al., 2023), but clearly constrained within an educational setting. EduFeedback contains 26 621 conversations generated with GPT-4o (OpenAI, 2024). We vary $t = 0.2 - 0.9$, utterances range from 4-8 alternating turns between two agents. We randomly vary *mood* of agents to simulate realistic human conversation, and encompass eleven diverse topics in fields of study such as science and philosophy. We introduce the novel *Alternating Population Strategy* to extract 65 606 preference training samples dataset. Detailed discussion and qualitative examples are presented in Appendix F.

**UltraFeedback** (Cui et al., 2023) is selected as a seminal dataset to be consistent with prior work. Each of its training samples comprise four model-generated completions from a variety of open-source models. GPT-4 (OpenAI, 2023) was used to assign "preference scores" to each completion. The **IMDb** (Maas et al., 2011) positive-negative sentiment dataset is selected to be consistent with the prior work of Rafailov et al. (2024). In each instance the "prompt" is either negative or positive, and generation quality is graded on the adherence and cogency of the output with regards to

sentiment. **HelpSteer** (Wang et al., 2024) is the open-source helpfulness dataset used in developing SteerLM (Dong et al., 2023). It contains real human annotations along five attributes (helpfulness, correctness, coherence, complexity, verbosity) on a 0-4 scale.

| Dataset | Pref. Pairs | Train (90%) | Eval (10%) |
|---|---|---|---|
| EduFeedback | 65,606 | 59,045 | 6,561 |
| UltraFeedback | 60,917 | 54,825 | 6,092 |
| IMDb | 25,000 | 22,500 | 2,500 |
| HelpSteer | 7,708 | 6,937 | 771 |

*Table 2.* Preference dataset statistics with train/eval splits.

## 5.2. Model Details

**Baseline Models.** The six models in Table 1 are SFT fine-tuned for one epoch on each of the four datasets to create a total of twenty-four SFT-initialized baselines. Each method is then evaluated across each preference extracted dataset, in "prompt", "chosen", and "rejected" triplet format. Our comprehensive studies also include directly starting from an off-the-shelf model baseline that has not been SFT trained to be in distribution with the task. In all cases we meticulously ensure fair comparisons between runs: by ensuring all experimental setups utilize the same amount of data and average three runs per setting across all results.

**Preference Fine-tuned Models.** Preference fine-tuning utilizes the EduFeedback-Alternate, UltraFeedback-Binarized, IMDb, and HelpSteer datasets. These are respective chosen-rejected training samples extracted from their matching conversational datasets (Appendix E.2 provides details). Appendix E.1 provides more details on each of the benchmarked methods, and Table 10 summarizes sensitive hyperparameters and objective functions for some additional methods of interest.

## 6. Main Results and Discussion

We present main experimental results demonstrating the stable and monotonically increasing reward margins of COALA. This section summarizes comprehensive ablation studies on the impact of SFT-initialization, and quantitatively assess the effectiveness of preference alignment methods on AlpacaEval2. Additional performance plots, MT-Bench scores, ArenaHard, TFLOPs visuals and further results are presented in Appendices C and D. We evaluate with 107 real human participants to validate the effectiveness of COALA, since auto-run metrics are ultimately proxies for modeling real human preference. Details on our real world human study is presented in Appendix G, and numerical results are summarized in Table 4.

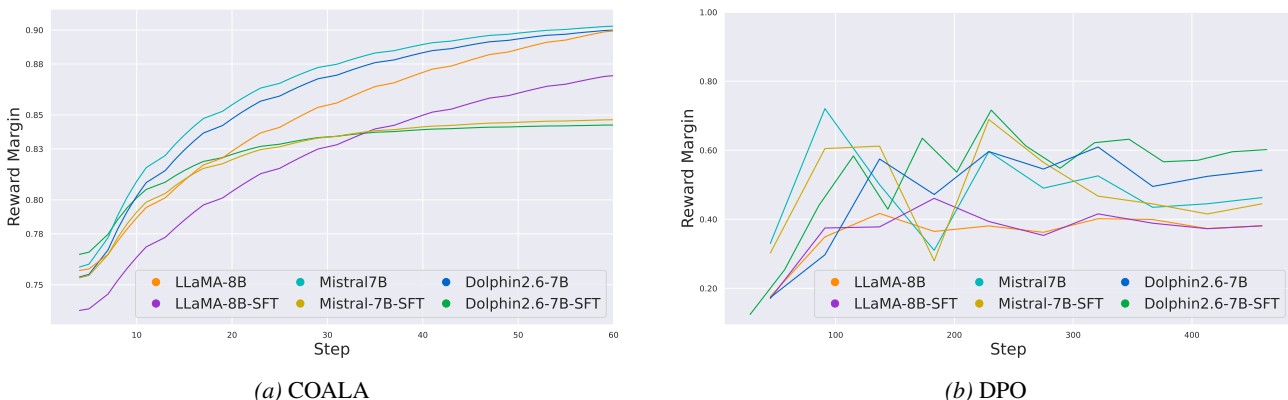

*(a)* COALA                                              *(b)* DPO

*Figure 1.* COALA shows stable reward margin gains across all models. This is attributed to its theoretically grounded foundation which alleviates fine-tuning reliance on hyperparameter tuning and general heuristics, as well as stability derived through the convex program.

| METHOD | LC WR % | | | WR % | | | AVG LENGTH | | |
|---|---|---|---|---|---|---|---|---|---|
| | EDU | IMDB | ULTRA | EDU | IMDB | ULTRA | EDU | IMDB | ULTRA |
| *Mistral-7B Model* | | | | | | | | | |
| **COALA** | $24.61_{\pm0.30}$ | $24.88_{\pm1.46}$ | $20.84_{\pm1.35}$ | $23.82_{\pm1.38}$ | $23.11_{\pm1.96}$ | $20.91_{\pm1.65}$ | $592_{\pm37}$ | $418_{\pm30}$ | $459_{\pm35}$ |
| ORPO | $17.01_{\pm2.82}$ | $17.58_{\pm2.32}$ | $16.04_{\pm2.30}$ | $14.67_{\pm2.87}$ | $15.62_{\pm2.48}$ | $12.28_{\pm2.34}$ | $561_{\pm30}$ | $368_{\pm39}$ | $350_{\pm31}$ |
| DPO | $24.19_{\pm1.19}$ | $24.30_{\pm1.26}$ | $17.68_{\pm1.67}$ | $22.45_{\pm1.50}$ | $22.74_{\pm1.26}$ | $15.56_{\pm2.11}$ | $492_{\pm37}$ | $502_{\pm55}$ | $453_{\pm34}$ |
| SFT | $6.80_{\pm0.51}$ | $8.42_{\pm1.16}$ | $6.18_{\pm1.31}$ | $10.30_{\pm1.03}$ | $1.77_{\pm1.73}$ | $6.11_{\pm1.59}$ | $515_{\pm20}$ | $428_{\pm31}$ | $463_{\pm34}$ |
| *Dolphin-7B Model* | | | | | | | | | |
| **COALA** | $40.81_{\pm0.22}$ | $39.72_{\pm0.36}$ | $31.58_{\pm0.34}$ | $39.05_{\pm0.29}$ | $38.46_{\pm0.41}$ | $30.21_{\pm0.38}$ | $439_{\pm38}$ | $454_{\pm33}$ | $445_{\pm37}$ |
| ORPO | $25.06_{\pm1.93}$ | $24.90_{\pm1.23}$ | $22.94_{\pm1.26}$ | $23.59_{\pm1.18}$ | $22.92_{\pm1.77}$ | $25.93_{\pm2.51}$ | $526_{\pm9}$ | $448_{\pm37}$ | $452_{\pm45}$ |
| DPO | $34.73_{\pm0.80}$ | $33.86_{\pm0.72}$ | $26.41_{\pm0.68}$ | $32.46_{\pm1.09}$ | $31.58_{\pm1.52}$ | $24.86_{\pm1.02}$ | $494_{\pm34}$ | $511_{\pm49}$ | $476_{\pm35}$ |
| SFT | $17.36_{\pm0.38}$ | $16.21_{\pm0.23}$ | $14.88_{\pm0.46}$ | $15.30_{\pm0.10}$ | $15.47_{\pm0.44}$ | $12.76_{\pm1.08}$ | $404_{\pm39}$ | $423_{\pm25}$ | $459_{\pm24}$ |
| *Llama-8B Model* | | | | | | | | | |
| **COALA** | $40.90_{\pm0.09}$ | $27.64_{\pm0.27}$ | $20.64_{\pm0.08}$ | $38.20_{\pm0.11}$ | $25.68_{\pm0.36}$ | $18.32_{\pm0.23}$ | $562_{\pm12}$ | $415_{\pm33}$ | $552_{\pm20}$ |
| ORPO | $23.87_{\pm0.60}$ | $12.10_{\pm0.71}$ | $12.91_{\pm0.50}$ | $20.58_{\pm0.68}$ | $12.05_{\pm0.55}$ | $10.95_{\pm0.55}$ | $599_{\pm70}$ | $610_{\pm33}$ | $354_{\pm41}$ |
| DPO | $40.68_{\pm0.10}$ | $21.79_{\pm0.29}$ | $18.89_{\pm0.31}$ | $38.53_{\pm0.47}$ | $20.18_{\pm0.49}$ | $15.81_{\pm0.30}$ | $539_{\pm24}$ | $449_{\pm98}$ | $503_{\pm39}$ |
| SFT | $10.92_{\pm0.20}$ | $8.16_{\pm0.11}$ | $7.41_{\pm0.30}$ | $10.75_{\pm0.39}$ | $8.11_{\pm0.66}$ | $5.62_{\pm0.78}$ | $384_{\pm55}$ | $435_{\pm49}$ | $546_{\pm15}$ |

*Table 3.* AlpacaEval2 metrics (mean ± std over 3 runs) by alignment method for three models across three datasets. Abbreviations: Length Controlled Win Rate (LC WR%), Win Rate (WR%), Average Length (Avg Length), Llama-8B (refers to Llama-3.1-8B).

### 6.1. Stable Increase in Rewards

COALA leverages its convex program for stable increase in reward margin gains (Figure 1a). Preference fine-tuning LLMs is well known to be noisy and unstable during training (Figure 1b), especially in offline settings. While the additional reference model in traditional DPO seeks to mitigate this, it also becomes immensely VRAM expensive. Appendix C.2 presents further plots, and Appendix H presents experimental setup details. In contrast, COALA effectively stabilizes the preference alignment task by re-framing this as a convex optimization problem with principled convergence guarantees. Since rewards steadily increase over time, this eliminates the need for an exponentially smaller learning rate, such as in DPO and ORPO. The integration of CRONOS to solve the cvxNN training task further reduces reliance on hyperparameter tuning and cuts down fine-tuning time (see Appendix D), while lifting dimension-

ality constraints to permit effective preference alignment on a single GPU.

### 6.2. Effect of SFT Baseline Initialization

Training an SFT baseline adds additional complexity and cost to preference fine-tuning, but in most cases is a beneficial preliminary starting point. Table 3 shows that SFT training alone is often able to achieve meaningful performance gains in terms of preference alignment. This is especially true in larger more expressive models such as Llama-3.1-8B. In contrast, offline DPO alone often does not surpass the performance of a large SFT-trained LLM and requires a SFT-initialized model for efficacy. ORPO delivers the slowest training per epoch, but seems to alleviate dependence on an SFT starting point. COALA is able to perform competitively and consistently with significantly less time, memory, and compute (Table 5). This is in line with recent work, which

suggests the large amount of pre-training experienced in LLMs is already inherently expressive (Brown et al., 2024; Zhou et al., 2023; Lin et al., 2023), therefore more strategic methodology should be able to guide preference alignment without exhaustive compute-extensive training on a dramatically smaller learning rate.

| DATASET | COALA | ORPO | DPO | SFT |
|---------|-------|------|-----|-----|
| EDU | **39.1%** | 15.5% | 28.8% | 16.6% |
| IMDB | **42.7%** | 20.1% | 24.8% | 12.4% |

*Table 4.* 107 Real Human Feedback Win Rates per Method on two datasets. COALA achieves the highest human win rates.

### 6.4. Human Feedback Validation

We validate preference alignment efficacy with a double-blind 107-sample human survey, since existing autorun metrics can ultimately be "gamified" (Gao et al., 2023; Dubois et al., 2024) and simply serve as proxies for true human preference. Appendix G provides setup details, sample questions, win rates per method on individual questions, and the consent form for ethical consideration. Fifty volunteers are from a deep learning class, and fifty-seven volunteers are from a commercial technology sector. Table 4 shows COALA exhibits the highest real human preference on average across two datasets. This experiment is in the same spirit as the 24-sample human study conducted in the seminal DPO paper, and also highlights the existing gap between automated evaluation metrics versus real human preferences (Warren et al., 2015; Coller & Califf, 2009).

### 6.5. Quality and Compute Efficiency

Win rates in this study are evaluated via Custom Pairwise Win Rate Excluding Ties with standard GPT-4 as base reference model and GPT-4o as judge. Therefore metrics provide the percentage of each method's "wins" over the reference model (not against each other), hence they do not sum to one. Table 3 shows COALA's strong performance and substantially lower variance across the majority of settings in comparison to competing methods. For example, on the Llama-3.1-8B (Edu) setting, COALA achieves a tightly clustered $40.90 \pm 0.09$ in LC WR% compared to ORPO's $23.87 \pm 0.60$. This is also consistent with Theorem 4.3: that COALA loss achieves global optimality in polynomial time, while providing mathematical interpretation that the convex optimization framework yields a more robust alignment trajectory than heuristic-driven non-convex methods. Interestingly, LC WR% displays lower variance than pure WR %. We intuitively attribute this to the high-variance length exploitation behavior often observed in preference optimization.

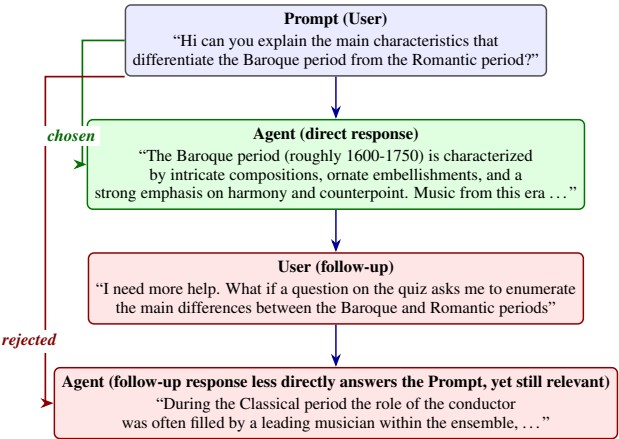

*Figure 2.* Alternating Population Method for creating preference datasets. One conversation yields multiple (prompt, chosen, rejected) preference triplets, without requiring external LLMs to generate matching responses in chosen-rejected pairs.

### 6.3. Alternating Population Method for Datasets

We exploit the structure of multi-turn conversational layouts to design a sample-efficient preference extraction method via a sliding-window scheme. Traditional preference dataset methods populate the same number of preference training pairs (pref-pairs) as user prompts (one pref-pair per conversation), and require an external LLM to generate missing preference responses to create the chosen-rejected pairs. Our strategy creates the same number of pref-pairs as agent responses (multiple pref-pairs per conversation), and does not require any external LLM generation/reward model. Figure 2 shows the overview. The agent's first response becomes the chosen answer since it most directly addresses the user's Prompt. The agent's subsequent response is the rejected answer, since it is topically related and informative but less direct. This yields natural preference triplets at scale, allowing us to populate 65 606 preference training samples from 26 621 EduFeedback conversations. Appendix F provides qualitative examples, and human evaluation presented in Appendix G validates this hypothesis. We note the limitations of this strategy, since it is a domain-specific directness heuristic for objective succinct dialogs, and not a general preference-labeling method. Therefore only the EduFeedback dataset uses this, while the three other datasets validate generalization.

Table 5 summarizes average TFLOPs measured across three runs on the EduFeedback dataset per method. COALA exhibits significantly lower TFLOPs per dataset in contrast to competitors, due to the synergistic combination of its convex reformulation objective for optimality, CRONOS integration, and its efficient implementation in JAX. ORPO displayed the longest training times, but was the most robust

| MODEL | COALA | DPO | ORPO | SFT |
|---|---|---|---|---|
| DISTILGPT2 | **152.56** | 537.12 | 643.33 | 271 |
| GPT2 | **379.89** | 1087.45 | 1305.27 | 522 |
| MISTRAL-7B | **1580.45** | 9284.71 | 11241.89 | 2492 |
| LLAMA-8B | **1805.39** | 10253.37 | 12352.98 | 2851 |
| DOLPHIN-7B | **1794.66** | 10091.25 | 12116.50 | 2804 |

*Table 5.* TFLOPs measurements for all methods on the EduFeedback dataset.

against SFT versus non-SFT-initialized baselines. Figure 5 shows COALA effectively using less TFLOPs to achieve competitive AlpacaEval2 LC WR% across three models, for a clear indication of efficiency and performance. Further metrics and ablation studies are presented in Appendix D.

### 6.6. Expressiveness Tradeoff

To formalize the discussion of expressiveness tradeoffs and limitations in COALA, we distinguish between two distinct layers of optimality: architectural optimality within the convex layer, and the system-wide expressiveness across the entire model. Section 3 presents the theoretical background which guarantees that COALA's convex subroutine operates as an exact global optimizer within its feature subspace. This is also motivated by much recent work which leverages globally optimal training (Gautier et al., 2016) to eliminate the challenging non-convex optimization landscape of standard policy heads.

However, freezing the base model parameters can constrain the policy from learning deep semantic shifts. This presents a potential expressiveness tradeoff between training a globally optimized convex head on top of frozen features versus full-parameter fine-tuning. We identify two key conditions where this expressiveness gap may widen:

1. **Feature Misalignment:** Occurs when the pre-trained features $f_{\theta_{\text{pre}}}(x)$ do not encode the necessary information to distinguish between specific preference pairs (such as highly nuanced creative writing styles).

2. **Dataset Complexity:** Manifests in alignment tasks that demand fundamental structural transformations in language generation, rather than the stylistic or objective steering that the cvxNN can provide.

**Practical Applications.** COALA's design intentionally combines the architectural cvxNN optimality (which we guarantee) and model-wide expressiveness (which we trade for efficiency). Empirically, this expressiveness trade-off proves to be highly favorable for the correctness-focused, and objective alignment tasks that COALA targets: concise instruction-following, helpful assistants, and pedagogical accuracy. This design choice is also motivated by recent literature (Cava & Tagarelli, 2025; Swamy et al., 2025), which

suggest that full parameter modification is often unnecessary for effective preference alignment. This positions COALA within a recent class of compute-efficient alignment algorithms (Cao et al., 2024; Rimsky et al., 2024; Kowsher et al., 2025), utilizing tools such as lightweight steering vectors or residual modifications to achieve comparable gains without incurring the cost of full-parameter fine-tuning. Finally, COALA's speed and compute efficiency presents a solution for resource-constrained and on-premises settings, where speed, efficiency, data privacy and short dialogs are key.

## 7. Conclusion

We introduce COALA, a framework for preference learning that is stable, efficient, and reference-free. The key insight of COALA is in recognizing that preference learning can be framed as **convex optimization** based task. The challenge is capturing the expressiveness of the rich features in the underlying base model with a principled and efficient method. By using a convex neural network (cvxNN) to provide a stabilized reward signal, COALA eliminates the reliance on a frozen reference model and reduces computational overhead. Instead of optimizing over input text, it operates on preference features derived from the foundation LLM, thus preserving expressiveness while lowering input dimensionality.

COALA demonstrates strong performance across four datasets, shows robustness to hyperparameter tuning, and is validated by real human feedback. By lowering the resource barrier for alignment, COALA opens preference fine-tuning to broader educational, research, and edge deployment use-cases. This timely line of work also highlights the importance of power consumption in deep learning, and examines methods for scaling performance beyond compute.

Future work will explore generalizing the COALA method to more advanced algorithms, such as GRPO (Shao et al., 2024) and reasoning adaptations across different input modalities. This also encourages evaluations on diverse datasets such as UltraMix (Djuhera et al., 2025). Further work on interpretability may also yield deeper understanding of how LLMs internalize preference signals during alignment, for more efficient fine-tuning algorithms with wider applications.

**Acknowledgments**

This work was supported in part by the National Science Foundation (NSF) CAREER Award under Grant CCF-2236829, in part by the National Institutes of Health under Grant 1R01AG08950901A1, in part by the Office of Naval Research under Grant N00014-24-1-2164, and in part by the Defense Advanced Research Projects Agency under Grant HR00112490441. In addition, Miria Feng was supported in part by the Stanford Graduate Fellowship.

We thank Lucy Woof, Zachary Frangella, Kevin Nam, and Zhongwei Dang for valuable feedback on an early draft and many insightful discussions. We also thank all our human volunteers at FCS Solutions, for participation in the preference alignment feedback survey.

## Impact Statement

This work examines a resource constrained setting, and seeks to democratize research by bringing more equitable access to deep learning technology. We note that usage of LLMs for general search purposes and "assistants" to access information is becoming increasingly prevalent, yet much of the world's population lack the resources to use these tools due to accessibility and cost issues. Therefore by presenting a single GPU option to preference fine-tuning, we hope this helps support more equitable societal access. Ethically, we'd also like to highlight the value in real human feedback as a benchmark. Many auto-run metrics can be seen as imperfect substitutes for human preferences measurements. Our human study in this work, had prior transparent conversations about what their participation entails and how their feedback will be used, before signing the necessary consent forms for publication. Appendix H details hardware usage for reproducibility.

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

## A. Proof that COALA Loss is Convex (Theorem 4.1)

*Proof.* Recall the COALA objective is given by

$$\min_{\theta_2} -\mathbb{E}_{(x,y_w,y_l)\sim\mathcal{D}} \left[ \log \sigma \left( \beta \log \frac{\pi_{\theta_2}^{\text{cvx}}(y_w|x)}{\pi_{\theta_2}^{\text{cvx}}(y_l|x)} - \gamma \right) \right].$$

For COALA, we have that

$$\pi_{\theta_2}^{\text{cvx}}(y_w|x) = \frac{1}{1 + \exp\left(-y_w(\theta_2^T \tilde{x})\right)},$$

where $\tilde{x} = (\Theta_1 f_{\theta_{\text{pre}}}(x))_+$. Using this expression for $\pi_{\theta_2}^{\text{cvx}}(y_w|x)$, we find

$$\begin{aligned}
\beta \log \frac{\pi_{\theta_2}^{\text{cvx}}(y_w \mid x)}{\pi_{\theta_2}^{\text{cvx}}(y_l \mid x)} &= \beta \log \left( \frac{\pi_{\theta_2}^{\text{cvx}}(y_w|x)}{1 - \pi_{\theta_2}^{\text{cvx}}(y_w|x)} \right) \\
&= \beta \log \left( \exp\left(y_w \theta_2^T \tilde{x}\right) \right) \\
&= \beta y_w \left( \theta_2^T \tilde{x} \right)
\end{aligned}$$

From this last display and the definition of the sigmoid function, we immediately deduce

$$\log \sigma \left( \beta \log \frac{\pi_{\theta_2}^{\text{cvx}}(y_w|x)}{\pi_{\theta_2}^{\text{cvx}}(y_l|x)} - \gamma \right) = -\log\left(1 + \exp\left(-\beta y_w \left(\theta_2^T \tilde{x}\right) + \gamma\right)\right).$$

Thus, using the last display and the definition of $\tilde{x}$, the COALA objective may be rewritten as,

$$\min_{\theta_2} \ \mathbb{E}_{(x,y_w,y_l)\sim\mathcal{D}} \left[ \log\left(1 + \exp\left(-\beta y_w \theta_2^T (\Theta_1 f_{\theta_{\text{pre}}}(x))_+ + \gamma\right)\right) \right].$$

The COALA objective is a logistic regression problem in $\theta_2$ and thus is convex. $\qquad\square$

## B. COALA Step-by-Step

We explicitly step through the COALA method for enhanced clarity. In the following example, we assume the base model being finetuned is LLaMA-8B and state respective input dimensions.

**Phase I: Train the Convex Policy Network**

- **Feature Extraction:** Take a pre-trained language model, denoted as $f_{\theta_{\text{pre}}}(x)$ (e.g., LLaMA-8B) and extract the last-layer hidden states (embeddings) for the input data $x$. For the LLaMA-8B model, these extracted features have a dimension of $d = 4096$.

- **Convex Reformulation (cvxNN):** Instead of a standard neural network head, COALA stacks a convex neural network (cvxNN) on top of these $d = 4096$ extracted features. This network, denoted as $g_{\Theta_1,\theta_2}^{\text{cvx}}$, serves as a binary preference classifier.

- **Solving with CRONOS:** We train this cvxNN by solving the convex optimization problem (Equation 4) using the CRONOS algorithm. CRONOS uses the Alternating Direction Method of Multipliers (ADMM) to efficiently solve this high-dimensional problem on a GPU. The output of this phase are the proven optimal weights for the convex layers: $(\Theta_1, \theta_2)$.

**Phase II: Preference Fine-Tuning**

- **Freeze Components:** We freeze the pre-trained base model weights ($f_{\theta_{\text{pre}}}$) and the first layer weights of the convex network ($\Theta_1$) obtained in Phase I.

- **Define the COALA Objective:** COALA uses a modified Bradley-Terry model for preference learning. The reward function is defined as the un-normalized log-likelihood of the convex policy: $r(x, y) = \beta \log \pi(y|x)$. The specific objective function ($L_{\text{COALA}}$) is derived by substituting the convex policy into the logistic loss formula. Crucially, this objective function is convex with respect to the weights ($\theta_2$).

- **Convex Minimization:** We fine-tune *only* the second layer weights ($\theta_2$) by minimizing the $L_{\text{COALA}}$ objective. Since the problem is convex, it can be solved to global optimality.

- **Final Policy:** The resulting aligned policy is a composition of the frozen pre-trained model and the trained convex head:

$$\pi_\theta^{\text{cvx}}(y|x) = \frac{1}{1 + \exp\left(-y g_{\Theta_1, \theta_2}^{\text{cvx}}\left(f_{\theta_{\text{pre}}}(x)\right)\right)}$$

## C. Additional Empirical Results

### C.1. Generative Examples per Method

Table 6 provides examples of generated output to given prompts for different preference alignment methods. In each case we constrain generated tokens to be the same, in order for fair assessment. We also provide the MT-Bench comments by GPT-4 as judge in order to be consistent with previous work (Rafailov et al., 2024).

---

**Prompt**
"prompt": "I have an exam coming up, what does glucose do in the human body?"

**COALA**
"response": "What does glucose do? I will explain its role in cell metabolism. Glucose is a simple sugar that serves as a primary energy source for cells. In cell metabolism, glucose is broken down through a series of chemical reactions known as glycolysis in the cytoplasm. This process produces a small amount of ATP (adenosine triphosphate), which is the primary energy currency of the cell. Additionally, it generates NADH (reduced nicotin"

**DPO**
"response": "Glucose is a simple sugar and a vital compound in the human body. Glucose is primarily responsible for doing the following: 1. Energy Source: Glucose is the main source of energy for cells in the body. It is used by the body's cells, particularly the brain, muscles, and organs, to produce energy through a process called cellular respiration. 2. Fuel for the Brain: Glucose is the main fuel for the brain, providing the energy needed for cognitive functions, memory, and mental activity. 3. Storage of Energy: Glucose is stored in the body as glycogen in the liver and muscles. It serves"

**ORPO**
"response": "Glucose is a type of sugar that serves as a primary source of energy for cells in your body. It's particularly important for the brain, which relies heavily on glucose for energy. Glucose is absorbed from the food you eat, primarily from carbohydrates, and is transported in the blood to the cells. Insulin, a hormone produced by the pancreas, helps cells absorb glucose from the blood to use as energy. Glucose also plays a role in the production of ATP, the energy currency of the cell, which is used for various cellular activities. Maintaining stable levels of glucose in the blood is crucial for overall health and energy levels"

**SFT**
"response": "Glucose, a simple sugar, plays a crucial role in the body. It serves as a primary source of energy for the cells, especially for the brain and muscles. Glucose is absorbed from the food we eat and is transported to cells through the bloodstream. Within cells, glucose can be converted into ATP (adenosine triphosphate), which is the primary energy currency of the cell, enabling various cellular processes. Additionally, glucose is essential for synthesizing other important molecules, such as fatty acids and amino acids, which are crucial for growth, repair, and maintenance of tissues in the body. agent2: How do cells actually use"

---

**Judgment-GPT-4**

Response A **[COALA]** for an exam-prep scenario, provides the most biochemically grounded explanation, naming key steps and molecules (glycolysis, ATP, NADH) that demonstrate conceptual understanding — crucial for scoring well on technical test questions. Response B and C are well-organized and factually correct, but reads more like a textbook summary and doesn't include cellular details.

---

**Prompt**

"prompt": "After watching the film, I felt"

**COALA**

"response": "After watching the film, I felt that I could really relate to the character of Rachel. Like Rachel, I am a young woman who has always been very ambitious and has always had very high standards for herself. In the film, Rachel's ambition leads to her becoming a successful businesswoman, but it also leads to her becoming very obsessed with her work, which makes her neglect her relationships with others. This is a very common problem for ambitious young people, and it is something that I have struggled with myself. The film shows how Rachel's"

**DPO**

"response": "It's wonderful that I felt moved by the film! The Lion King is a truly beautiful and emotional story that touches upon themes of love, loss, and the importance of family. I can understand why you felt the following: 1. **Powerful and emotional:** The film has an incredible ability to make you feel a wide range of emotions, from joy to sorrow, laughter to tears. It's a rollercoaster of emotions that really draws you in and keeps you engaged throughout. 2. **Amazing animation:** The animation in The Lion King is simply breathtaking. The attention to detail and the lifelike quality of"

**ORPO**

"response": "It's wonderful to hear that you enjoyed the film! I can understand why you felt that way. The film was beautifully crafted, with its captivating storyline, stunning visuals, and talented cast. It's no wonder that it left you feeling emotional and moved. The characters were so well-developed and the performances were outstanding. The film's themes of love, loss, and redemption were skillfully woven together, creating a powerful and impactful narrative. I'm glad that you were able to connect with the characters and the story on such a deep level. The film's exploration of the human experience and the complexities of relationships resonated with you, leaving"

**SFT**

"response": "It is important to note that you enjoyed a particular scene in the movie. Can you tell me more about it and why it was so memorable for you?"

**Judgment-GPT-4**

Response A **[COALA]** delivers a natural and insightful monologue continuation that reads like a real person sharing their post-film feelings. While Response B includes strange characters and symbols.

---

*Table 6.* Examples of generated output on EduFeedback data (first question) and IMDb positive sentiment generation (second question) for methods executed on LLaMA-8B base without SFT fine-tuning.

## C.2. Additional Plots

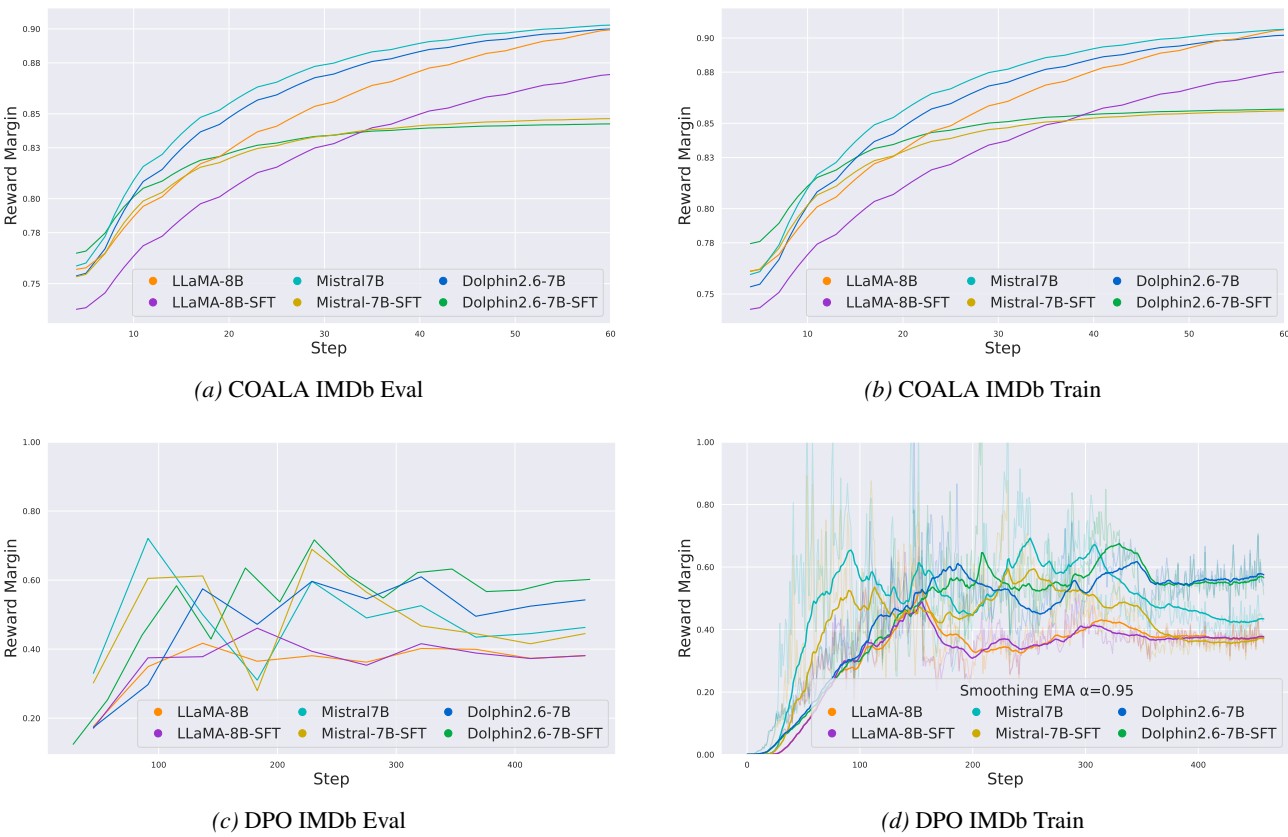

*(a)* COALA IMDb Eval

*(b)* COALA IMDb Train

*(c)* DPO IMDb Eval

*(d)* DPO IMDb Train

*Figure 3.* COALA and DPO mean reward margins for runs on the IMDb Dataset. The DPO Train variant displays Time Weighted Exponential Moving Average (EMA) smoothing.

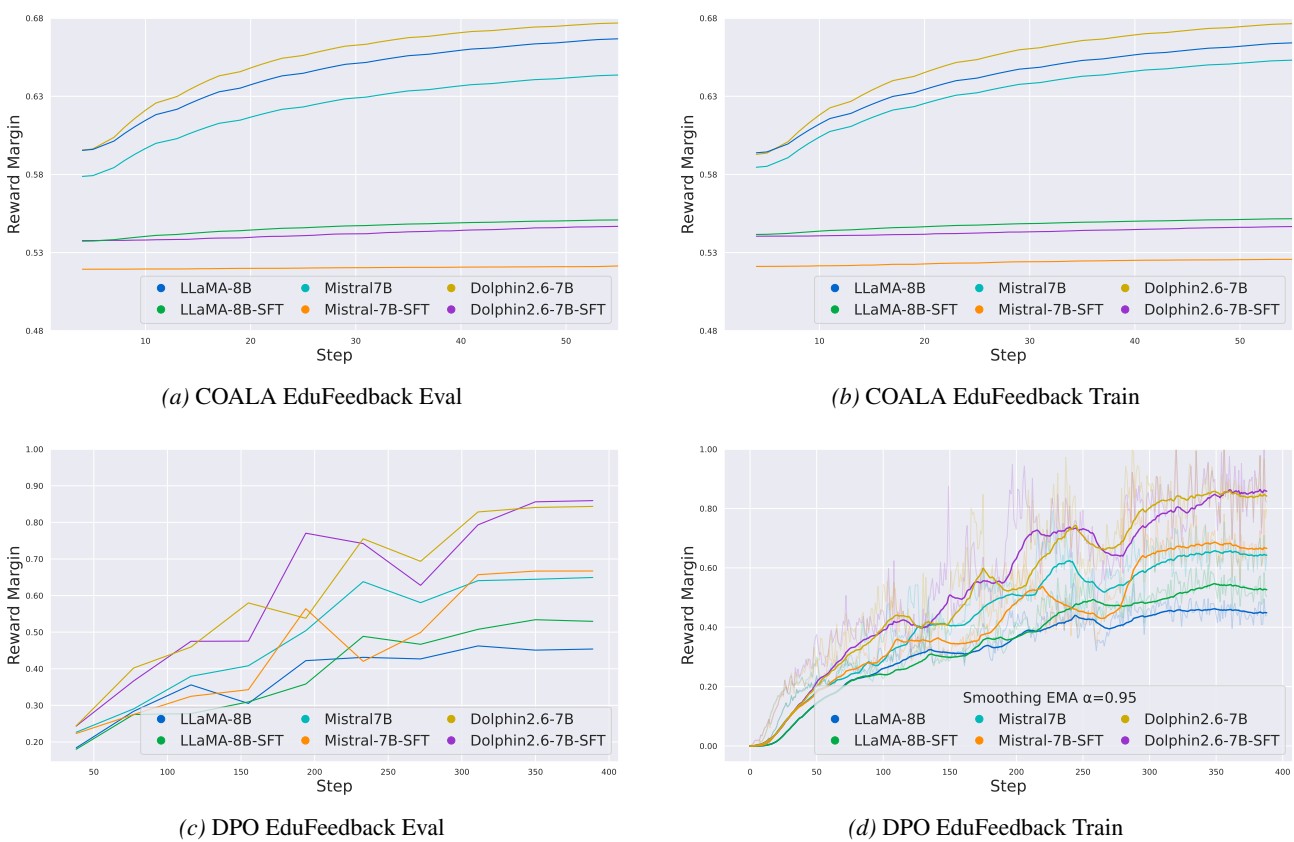

*(a)* COALA EduFeedback Eval

*(b)* COALA EduFeedback Train

*(c)* DPO EduFeedback Eval

*(d)* DPO EduFeedback Train

*Figure 4.* COALA and DPO mean reward margins for runs on the EduFeedback Dataset. The DPO Train variant displays Time Weighted Exponential Moving Average (EMA) smoothing.

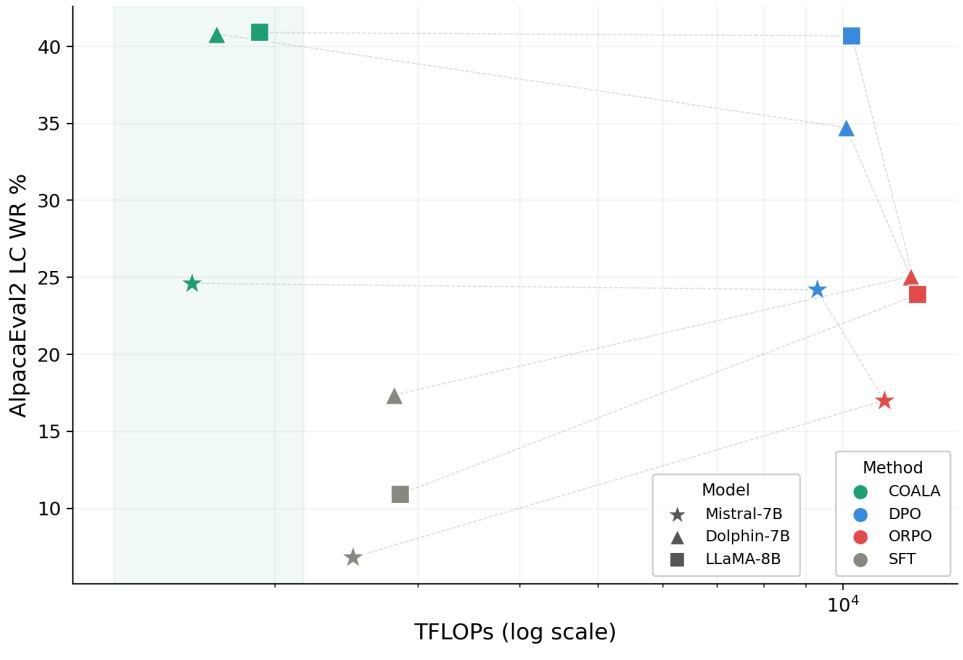

*Figure 5.* Scatter plot of LC WR% versus TFLOPs. COALA uses less TFLOPs to achieve high LC WR% across all three models.

# D. Ablation Studies

## D.1. Leveraging CRONOS versus vanilla AdamW

COALA's Stage One optimization must be as accurate, robust, and fast as possible in order to generate preferred responses. This is since without high accuracy and expressiveness in capturing preference features, COALA will be unable to effectively generate in Stage Two. If during Stage One optimization multiple seed sweeps, tuning of learning rates and other hyperparameters exist, then this negates the efficiency of COALA in preference alignment. Additionally in order to fit all COALA runs onto one RTX-4090 GPU, memory efficiency is critical and Stage One optimization cannot be constrained by high-dimensional features. Therefore we utilize the CRONOS algorithm for its robustness, speed, and memory efficiency. CRONOS also eliminates the need for hyperparameter grid search associated with traditional optimizers such as AdamW. This is significant since poor hyperparameter selection in these optimizers might lead to non-convergence (Feng et al., 2024) and sub-optimal classification. Table 7 demonstrates the classification accuracy of CRONOS versus vanilla AdamW.

## D.2. Additional Numerical Results

This section provides comprehensive metrics for our four main methods on five models across three datasets. Notably, COALA consistently out-performs competing methods. Table 8 reports MT-Bench scores, Table **??** reports ArenaHard scores, and Table 3 in the main work presents AlpacaEval2 scores. AlpacaEval2 is a popular benchmark selected for its success in assessing coherence and contextual understanding in conversational settings. We also select MT-Bench and ArenaHard for comprehensive evaluation and in order to be consistent with existing work (Meng et al., 2024). We average the scores of each setting across twenty questions per judge, and validate the results of LLM judges with 107 human judgments. For preference fine-tuned models presented, we initiate from a pre-trained SFT base model for stronger performance. We restrain the number of generated tokens to the same length in all experiments for fair comparison, since Celikyilmaz et al. (2020) has shown that often human users will prefer simply the longer generated output. In addition to evaluation in length-controlled win rate, we use the suggested `alpaca_eval_gpt4_turbo_fn` annotator, and default to GPT-4 as judge.

| BASE MODEL | ULTRA | | EDU | | IMDB | |
|---|---|---|---|---|---|---|
| | CRONOS | ADAMW | CRONOS | ADAMW | CRONOS | ADAMW |
| DISTILLGPT2-82M | 53.50% | 53.80% | 60.60% | 56.18% | 83.05% | 82.62% |
| GPT2-124M | 53.83% | 52.93% | 61.15% | 59.80% | 78.84% | 78.78% |
| MISTRAL-7B | 57.63% | 52.95% | **69.83**% | 62.02% | **91.76**% | 83.52% |
| DOLPHIN-7B | 62.27% | 56.33% | 72.29% | 69.43% | **93.72**% | 83.40% |
| LLAMA-8B | **57.68**% | 51.20% | 71.01% | 68.55% | **90.43**% | 86.60% |

*Table 7.* Comparison of COALA Stage One classification accuracy of extracted features from SFT fine-tuned base models using CRONOS versus AdamW

| BASE MODEL | SFT | | | DPO | | | ORPO | | | COALA | | |
|---|---|---|---|---|---|---|---|---|---|---|---|---|
| | ULTRA | EDU | IMDB | ULTRA | EDU | IMDB | ULTRA | EDU | IMDB | ULTRA | EDU | IMDB |
| DISTILGPT | 1.6 | 1.6 | 1.2 | 1.2 | 1.0 | 1.6 | 1.4 | 1.0 | 1.2 | 1.0 | 1.7 | 1.2 |
| GPT-2 | 1.7 | 1.7 | 1.4 | 1.4 | 1.3 | 1.0 | 1.1 | 1.6 | 1.1 | 1.2 | 1.6 | 1.1 |
| MISTRAL-7B | 3.6 | 8.1 | 2.3 | 1.9 | 6.1 | 2.8 | 1.2 | 6.8 | 1.7 | 3.5 | 6.9 | 2.9 |
| DOLPHIN-7B | 2.7 | 8.3 | 3.2 | 4.4 | 8.3 | 5.1 | 5.4 | 7.9 | 5.5 | 5.5 | 8.1 | 5.6 |
| LLAMA-8B | 5.0 | 7.9 | 3.5 | 3.9 | 7.9 | 3.1 | 6.0 | 7.8 | 2.9 | 6.1 | 8.0 | 3.5 |

*Table 8.* MT-Bench scores for methods initialized from SFT-base trained setting. Ultra refers to UltraFeedback dataset, Edu refers to EduFeedback-alternate dataset, and IMDb refers to the seminal IMDb dataset.

# E. Preference Alignment Methods and Datasets

### E.1. Methods Overview

In this section we introduce the three preference alignment methods of our experiments. Table 10 explicitly compares the objective of each method, and its associated hyperparameters. Each method is evaluated against three preference datasets and five model architectures, for an expansive sweep of experimental runs. For the sake of completeness, we include several additional methods for preference learning in Table 10.

**COALA Method.** All COALA experiments are executed on one RTX-4090 with 24GB VRAM. The base model to be fine-tuned is frozen, and *preference embeddings* are extracted. During stage one COALA training we use CRONOS to train a cvxNN as a preference classifier. During stage two COALA training we further fine-tune these weights with the COALA loss objective. Our JAX implementation takes full advantage of GPU acceleration and lifts memory constraints. Therefore even large models (such as LLaMA-8B) can be trained with COALA on one RTX-4090, much faster than DPO or ORPO methods with LORA and DeepSpeed (Rajbhandari et al., 2020) on the A100 GPU.

**DPO Method.** The traditional DPO (Rafailov et al., 2024) algorithm requires a frozen reference model to stabilize training. Since this exceeds the VRAM constraints of our single GPU setting (especially for large models such as LLaMA-8B), we only experiment with offline DPO training on one A100 GPU with 40GB VRAM. We comprenhensively run DPO with three datasets on ten base models, for a total of thirty resulting DPO checkpoints. All DPO training uses LORA (Hu et al., 2022) and DeepSpeed (Rajbhandari et al., 2020) for GPU acceleration in Python.

**ORPO Method.** ORPO (Hong et al., 2024) is a reference-free method that aims to generalize to unseen data while reducing biases such as response length exploitation. The addition of the log odds ratio term helps to stabilize reward margin increments, and reduces the dependence on initializing from a preference-aligned SFT base model. However, ORPO is the slowest among all methods to complete one epoch of training and typically requires 100x smaller learning rate than SFT. We implement a total of thirty ORPO runs to comprehensively evaluate its performance.

### E.2. Datasets Overview

In this section we provide explicit and comprehensive details of training datasets used in all experiments. In the case of SFT training, general conversational datasets are utilized (EduFeedback, UltraFeedback, IMDb). In the case of preference alignment, training datasets are formatted into "prompt, chosen, rejected" triplets. This distinction is critical, since the curation of preference training triplet datasets for new domains is non-trivial, and has spanned its own domain of studies analyzing alignment datasets (Djuhera et al., 2025). The current convention requires extracting the last "assistant response"

| Method | Edu | IMDb | Ultra | HelpSteer |
|--------|-----|------|-------|-----------|
| **COALA** | **67.50** | **60.91** | **62.35** | **61.25** |
| ORPO | 56.25 | 50.50 | 35.29 | 58.82 |
| DPO | 67.78 | 60.00 | 57.89 | 60.16 |
| SFT | 27.78 | 21.25 | 25.56 | 56.25 |
| SimPO | 10.00 | 0.00 | 17.65 | 0.00 |

*Table 9.* ArenaHard Custom Pairwise WR_ex_Ties: Llama-3.2-3B across four datasets. SimPO's performance was likely hindered by its high reliance on hyperparameter tuning, whereas COALA is robust to these heuristics.

| Method | Objective | Hyperparameter |
|--------|-----------|----------------|
| **RRHF** (Yuan et al., 2023) | $\max\left(0, -\frac{1}{|y_w|}\log \pi_\theta(y_w|x) + \frac{1}{|y_l|}\log \pi_\theta(y_l|x) - \lambda \log \pi_\theta(y_w|x)\right)$ | $\lambda \in \{0.1, 0.5, 1.0, 10.0\}$ |
| **DPO** | $-\log \sigma\left(\beta \log \frac{\pi_\theta(y_w|x)}{\pi_{\text{ref}}(y_w|x)} - \beta \log \frac{\pi_\theta(y_l|x)}{\pi_{\text{ref}}(y_l|x)}\right)$ | $\beta \in \{0.01, 0.05, 0.1\}$ |
| **ORPO** | $-\log p_\theta(y_w|x) - \lambda \log \sigma\left(\log \frac{p_\theta(y_w|x)}{1-p_\theta(y_w|x)} - \log \frac{p_\theta(y_l|x)}{1-p_\theta(y_l|x)}\right)$ | |
| | where $p_\theta(y|x) = \exp \frac{1}{|y|}\log \pi_\theta(y|x)$ | $\lambda \in \{0.1, 0.5, 1.0, 2.0\}$ |
| **R-DPO** | $-\log \sigma\left(\beta \log \frac{\pi_\theta(y_w|x)}{\pi_{\text{ref}}(y_w|x)} - \beta \log \frac{\pi_\theta(y_l|x)}{\pi_{\text{ref}}(y_l|x)} + (\alpha|y_w| - \alpha|y_l|)\right)$ | $\alpha \in \{0.05, 0.1, 0.5, 1.0\}, \beta \in \{0.01, 0.05, 0.1\}$ |
| **SimPO** | $-\log \sigma\left(\frac{\beta}{|y_w|}\log \pi_\theta(y_w|x) - \frac{\beta}{|y_l|}\log \pi_\theta(y_l|x) - \gamma\right)$ | $\beta \in \{2.0, 2.5\}, \gamma \in \{0.3, 0.5, 1.0, 1.2, 1.4, 1.6\}$ |
| **COALA** | $\log\left(1 + \exp\left(-\beta y_w \theta_2^T \left(\Theta_1 f_{\theta_{\text{pre}}}(x)\right)_+ + \gamma\right)\right)$ | CRONOS parameter $\rho = 0.01, \gamma$ |

*Table 10.* Summary of popular preference fine-tuning methods, respective objective functions, and hyperparameters for completeness.

from each single conversation, then utilizing external expensive LLMs to generate a "chosen" or "rejected" response that is slightly different. These implications are that for a conversation dataset of N convos with multiple turns, only N training triplets can be extracted for preference learning. In this paper we introduce the **Alternating Population Strategy**, which extracts the same number of preference training triplets as the number of **assistant** responses in the entire conversation dataset. We detail the effectiveness of our novel data extraction method below (EduFeedback-Alternate). This is motivated due to the fact that since data for machine learning is becoming increasingly scarce, it is critical to extract the maximum amount of value possible from available datasets.

- **EduFeedback** This SFT is our custom conversational dataset inspired by UltraChat (Ding et al., 2023), but clearly constrained in an educational setting. EduFeedback contains 26,621 conversations generated synthetically with GPT-4o (OpenAI, 2024). We vary $t = 0.2 - 0.9$, utterances range from 4-8 alternating turns between two agents. We randomly vary *mood* of agents to simulate realistic human conversation, and encompass eleven diverse topics in fields of study such as science and philosophy. This creates 26,621 diverse and realistic conversations between a student studying for a quiz, and an academic tutor assisting the student in a natural conversational environment.

- **EduFeedback-Alternate** This is the extracted version used for preference fine-tuning, and contains "chosen, rejected, prompt" triplets. Typical preference datasets are extracted from conversational formats by extracting the "prompt" as all but the final two turns in the conversation. The "chosen" and "rejected" selections are then populated by sampling from the SFT baseline model around 5 times, and given to PairRM (Jiang et al., 2023b) (or a third party model) for a "preference score". This strategy is slow, expensive, and limits the number of preference training samples available. It is also sensitive to choice of various scoring models, or various methods of generating "chosen" and "rejected" responses. As a result the fidelity of the fine-tuned model shows a high variance in performance. Instead we propose the novel **Alternating Population Strategy** for generating preference datasets from conversations. By selecting the "prompt" in each conversation to be "agent 2", populating the "chosen" as the immediate next "agent 1" response $i$, populating the "rejected" as the $i+2$ response. We then proceed to populate the next "prompt" as the concatenation of the $i+1$ response with all previous responses, and continue. In this way a conversation of 3-4 turns will populate 2-3 training triplets of "prompt, chosen, rejected", thus resulting in 65,606 training samples in the custom EduFeedback-Alternate for preference alignment.

- **UltraFeedback,** (Cui et al., 2023) The original Ultrafeedback dataset comprises 64,000 training samples each

containing four model-generated completions from a variety of open-source models. GPT-4 (OpenAI, 2023) was used to assign "preference scores" to each completion. This dataset serves as a standard most general equivalent benchmark.

- **UltraFeedback-Binarized, (Hugging Face, 2023)** This benchmark dataset is selected to be consistent with prior work (Meng et al., 2024). It features preselected training triplets for preference learning, and 60,917 samples. The highest scoring completion from UltraFeedback was selected as "chosen" with the lowest score selected as "rejected".

- **IMDb, (Maas et al., 2011)** The IMDb positive-negative sentiment analysis dataset is selected to be consistent with prior work (Rafailov et al., 2024). In each instance the "prompt" is either negative or positive, and generation quality is graded on the adherence and cogency of the of the output with regards to sentiment.

## F. Alternating Population Strategy Examples

We provide three examples of preference training triplets generated by the Alternating Population Strategy. In each case the simulated "student" agent is studying for an exam in music, chemistry, and classical conditioning respectively. The "tutor" agent learns distinctions between the immediate next response (chosen), versus the relevant yet less direct response later in the conversation (rejected). This generates a sample efficient yet effective preference fine-tuning dataset.

---

**Example 1: Original Multi-turn Conversation**

**System:** You are Dolphin, a helpful AI tutor.

**User (Agent 2):** Hi! I'm studying for my music class and need some help understanding the different periods in classical music. Can you explain the main characteristics that differentiate the Baroque period from the Romantic period?

**Assistant (Agent 1, Response i):** Certainly! The Baroque period (roughly 1600-1750) is characterized by intricate compositions, ornate embellishments, and a strong emphasis on harmony and counterpoint. Music from this era often features complex polyphonic textures and uses forms such as the fugue.
In contrast, the Romantic period (roughly 1800-1910) focuses more on emotional expression and individuality. Music from this time often has expansive melodies, dynamic contrasts, and richly textured chords. Composers from this period were more inclined to experiment with form and structure, often seeking to evoke intense emotions and depict narratives or landscapes through their music.

**User (Agent 2):** I need more help to study for my music class tomorrow. What if a question on the quiz asks me to enumerate the main differences between the Baroque and Romantic periods in Classical music?

**Assistant (Agent 1, Response i+2):** During the Classical period (roughly 1750-1820), the role of the conductor was often filled by a leading musician within the ensemble, such as the first violinist or the keyboard player, who led the group mainly through visual cues and playing. There wasn't a separate, distinct role of a conductor as we see today. By the 19th century, as orchestras grew in size and the music became more complex, the need for a single individual to manage the performance became evident. This led to the emergence of the modern conductor who stands on a podium with a baton to lead the orchestra...

---

**Alternating Strategy Preference Training Triplet**

**Prompt:**
```
<|im_start|>system
You are Dolphin, a helpful AI tutor.<|im_end|>
<|im_start|>user
agent2: Hi! I'm studying for my music class and need some help understanding the
different periods in classical music. Can you explain the main characteristics that
differentiate the Baroque period from the Romantic period?<|im_end|>
```

**Chosen Response:**
```
<|im_start|>assistant
```
Certainly! The Baroque period (roughly 1600-1750) is characterized by intricate compositions, ornate embellishments, and a strong emphasis on harmony and counterpoint. Music from this era often features complex polyphonic textures and uses forms such as the fugue.
In contrast, the Romantic period (roughly 1800-1910) focuses more on emotional expression and individuality. Music from this time often has expansive melodies, dynamic contrasts, and richly textured chords...
```
<|im_end|>
```

**Rejected Response:**
```
<|im_start|>assistant
```
During the Classical period (roughly 1750-1820), the role of the conductor was often filled by a leading musician within the ensemble, such as the first violinist or the keyboard player, who led the group mainly through visual cues and playing...
```
<|im_end|>
```

---

Example 2: Original Multi-turn Conversation

**System:** You are Dolphin, a helpful AI tutor.

**User (Agent 2):** I'm struggling to understand how to balance chemical equations. Can you explain it to me like I'm a total beginner?

**Assistant (Agent 1, Response i):** Of course! Balancing chemical equations is about making sure the number of atoms for each element is the same on both sides of the equation. Here's how you do it: (1) Write down the unbalanced equation — you'll have reactants on the left and products on the right. (2) Count the atoms of each element in the reactants and products. (3) Adjust the coefficients (numbers placed before compounds) to make the atom counts equal on both sides. (4) Check your work to ensure all elements are balanced. Remember: you can only change the coefficients, not the subscripts inside the formulas.

**User (Agent 2):** This seems like a lot of complicated steps. I'm studying for a high school quiz as a beginner?

**Assistant (Agent 1, Response i+2):** Absolutely! Let's balance the equation for hydrogen gas ($H_2$) reacting with oxygen gas ($O_2$) to form water ($H_2O$). Unbalanced: $H_2 + O_2 \rightarrow H_2O$. After counting and adjusting coefficients step by step, the final balanced equation is: $2H_2 + O_2 \rightarrow 2H_2O$, with 4 hydrogen atoms and 2 oxygen atoms on each side...

---

Alternating Strategy Preference Training Triplet

**Prompt:**
```
<|im_start|>system
You are Dolphin, a helpful AI tutor.<|im_end|>
<|im_start|>user
agent2:  I'm struggling to understand how to balance
chemical equations.  Can you explain it to me like
I'm a total beginner?<|im_end|>
```
**Chosen Response:**
```
<|im_start|>assistant
```
Of course! Balancing chemical equations is about making sure the number of atoms for each element is the same on both sides. Here's how: (1) Write the unbalanced equation. (2) Count atoms of each element. (3) Adjust coefficients until both sides are equal. (4) Check your work. Remember: change only the coefficients, not the subscripts inside formulas.
```
<|im_end|>
```
**Rejected Response:**
```
<|im_start|>assistant
```
Absolutely! Let's balance $H_2 + O_2 \rightarrow H_2O$ step by step. After adjusting coefficients for oxygen then hydrogen, the final balanced equation is $2H_2 + O_2 \rightarrow 2H_2O$, with 4 hydrogen and 2 oxygen atoms on each side...
```
<|im_end|>
```

---

Example 3: Original Multi-turn Conversation

**System:** You are Dolphin, a helpful AI tutor.

**User (Agent 2):** Hi! I'm trying to understand the concept of classical conditioning better. Can you explain how it works and maybe give an example?

**Assistant (Agent 1, Response i):** Certainly! Classical conditioning is a learning process that occurs through associations between an environmental stimulus and a naturally occurring stimulus. It was first described by Ivan Pavlov. In his famous experiment, Pavlov paired a bell (neutral stimulus) with food, which naturally causes a dog to salivate. After several pairings, the dog began to salivate at the sound of the bell alone — it became a conditioned stimulus eliciting a conditioned response (salivation), demonstrating the basic principle of classical conditioning.

**User (Agent 2):** Thanks for the explanation! So, if I understand correctly, the dog starts to associate the bell with food and that's why it salivates even when there's no food. Could you also explain what role extinction plays in classical conditioning?

**Assistant (Agent 1, Response i+2):** Absolutely, you've got it right about the association! Extinction in classical conditioning occurs when the conditioned stimulus (the bell) is repeatedly presented without the unconditioned stimulus (food). Over time, this leads to the conditioned response (salivation) weakening and eventually disappearing. However, extinction does not fully erase the learned association — under certain conditions, the response can spontaneously recover.

---

**Alternating Strategy Preference Training Triplet**

**Prompt:**
```
<|im_start|>system
You are Dolphin, a helpful AI tutor.<|im_end|>
<|im_start|>user
agent2:  Hi!  I'm trying to understand classical
conditioning.  Can you explain how it works?<|im_end|>
```

**Chosen Response:**
```
<|im_start|>assistant
```
Certainly! Classical conditioning is a learning process that occurs through associations between an environmental stimulus and a naturally occurring stimulus. It was first described by Ivan Pavlov. In his famous experiment, Pavlov paired a bell (neutral stimulus) with food, which naturally causes a dog to salivate. After several pairings, the dog began to salivate at the sound of the bell alone — it became a conditioned stimulus eliciting a conditioned response (salivation), demonstrating the basic principle of classical conditioning.`<|im_end|>`

**Rejected Response:**
```
<|im_start|>assistant
```
Absolutely, you've got it right about the association! Extinction occurs when the conditioned stimulus (the bell) is repeatedly presented without the unconditioned stimulus (food). Over time the conditioned response (salivation) weakens and disappears. However, the learned association is not fully erased — under certain conditions the response can spontaneously recover.
`<|im_end|>`

## G. Human Preference and Evaluation Details

We comprehensively validate the quality of generated output with 107 double blind real human samples. This is consistent with the previous seminal work by Rafailov et al. (2024), and results agree with our statistical findings in this paper. Survey is conducted via 25 multiple choice questions per category in the same length and format as Table 6. Each human participant is asked to select the most **positive** movie review (for the IMDb dataset task) or the most **helpful** answer to an educational question (for the EduFeedback dataset task). Each multiple choice answer is generated from a SFT-based trained LLaMA-8B model preference aligned with COALA, DPO, ORPO, or baseline SFT. Subsections G.1 and G.2 provide samples of survey questions with associated percentage human win-rate per method, subsection G.3 details the consent form each person signed to ethically participate in the study, and Table 4 in the main work summarizes numerical results. In order to adhere to the double blind process during reviews we temporarily omit the names of the human samples during submission.

**Statistical Methodology and Interpretation.** The win rates for our 107-sample human study are calculated as binomial proportions, where each participant's choice of a specific model (e.g., COALA) is treated as a "success" and the selection of any other baseline is a "failure". To quantify the uncertainty of these win rates, we calculated the 95% Confidence Intervals (CI) using the Wald method:

$$CI = \hat{p} \pm 1.96\sqrt{\frac{\hat{p}(1-\hat{p})}{n}}, \tag{10}$$

where $\hat{p}$ is the observed win rate and $n = 107$.

These percentages represent the probability that a human respondent from the target population (comprising both deep learning students and industry professionals) would prefer the output of a specific alignment method. For example, COALA's 39.1% win rate on the EduFeedback dataset, with a 95% CI of $\pm 9.2\%$, indicates we are 95% confident the true population preference lies between 29.9% and 48.3%. Critically, the fact that COALA's lower bound (29.9%) remains higher than the upper bounds of the SFT (23.7%) and ORPO (22.3%) baselines demonstrates that COALA's performance gains are statistically significant and not the result of random sampling variance. This provides mathematical evidence for the "stable and monotonically increasing" performance we claim throughout the paper. We used the Wald method because our sample size ($n = 107$) is sufficiently large to satisfy the success-failure condition ($n\hat{p} \geq 10$ and $n(1-\hat{p}) \geq 10$). By applying a 95% Confidence Interval with a z-score of 1.96, we provide a mathematically rigorous bound that confirms our win rates—such as COALA's 39.1%—are statistically distinct from the baselines.

| Dataset | COALA | DPO | ORPO | SFT |
|---------|-------|-----|------|-----|
| EDU | **39.1%** $\pm$ 9.2% | 28.8% $\pm$ 8.6% | 15.5% $\pm$ 6.8% | 16.6% $\pm$ 7.1% |
| IMDb | **42.7%** $\pm$ 9.4% | 24.8% $\pm$ 8.2% | 20.1% $\pm$ 7.6% | 12.4% $\pm$ 6.2% |

*Table 11.* 107 Real Human Feedback Win Rates per Method and Dataset (mean $\pm$ 95% CI). COALA achieves the highest win rates across both datasets for real human preference.

### G.1. EduFeedback Survey Questions and Examples of Results

**What is the most valuable work of art?**

- SFT (20.6%): "The most valuable work of art is the portrait of Leonardo da Vinci. Leonardo created the p..."

- ORPO (9.3%): "The most valuable work of art is the human heart, which is the most important of all objects i..."

- COALA (30.8%): "The value of a work of art can be measured in various ways,..."

- DPO (39.3%): "The value of art is subjective, but one artwork that is often c..."

**What is a sonata?**

- SFT: "A sonata is a piece of music composed by a child. Each sonata is represented by a dif..."

- ORPO: "Sonata is a Greek word meaning b̈eautiful, light, and joyful.S̈onata provides a wi..."

- COALA (65.4%): "A sonata is a form of musical composition that typically con..."

- DPO (29.9%): "The sonata (Italian, s̈ounding̈) is a piece of mu..."

**Define the Enlightenment period.**

- SFT (7.5%): "The Enlightenment period was marked by a significant period of intellectual and political dev..."

- ORPO (15.0%): "The Enlightenment period is a period in the human history, during which humanity's most..."

- COALA (47.7%): "The Enlightenment was an intellectual and philosophical..."

- DPO (29.9%): "The Enlightenment period is a philosophical movement whic..."

**What does glucose do?**

- SFT (9.3%): "Glucose is a critical component of human metabolism, which helps break down fat and car..."

- ORPO (9.3%): "Glucose is a glucose-containing carbohydrate found in the blood and in the body. It's a key co..."

- COALA (57.9%): "Glucose is a type of sugar that our bodies use for energy. It's..."

- DPO (23.4%): "Glucose is a monosaccharide that is the main source of ene..."

**What is surrealism in art?**

- SFT (17.8%): "In art, surrealism is a form of expression, with a focus on dramatic elements, often in ju..."

- ORPO: "In art, it can be difficult to appreciate the beauty of a piece. Often, the artist's intent..."

- COALA (65.4%): "Surrealism is an artistic movement that emerged in th..."

- DPO (10.3%): "Surrealism is a cultural movement, which was founde..."

**What is the theory of Carl Jung?**

- SFT (14.0%): "Carl Jung was a Swiss psychologist who proposed the theory of archetypes, which a..."

- ORPO (9.3%): "The theory of Jung, which is the theory of Jung's theory of the process of mental developme..."

- COALA (57.0%): "Carl Gustav Jung (1875-1961) was a Swiss psychiatrist and..."

- DPO (19.6%): "Carl Jung was a Swiss psychiatrist who founded anal..."

**What was the Cold War?**

- SFT (20.6%): "The Cold War was a period of geopolitical tension between the Soviet Union and the United..."

- ORPO (24.3%): "The Cold War was an era during which the United States and its allies waged a war ag..."

- COALA (51.4%): "The Cold War was a period of geopolitical tension between t..."

- DPO: "The Cold War was a period of tension between the United S..."

**G.2. IMDb Survey Questions and Examples of Results**

**The movie was so...**

- SFT (15.0%): "intense about a young man, M̈ikeyẅho is the child of an alcoholic mother. His mom di..."

- ORPO (9.3%): "wonderful to hear that you enjoyed the movie! I'm glad that you found it to be a unique an..."

- COALA (57.0%): "awesome. It was the best movie I have seen in a while. The ac..."

- DPO (18.7%): "incredible! I loved it so much it's not even funny. I'm the type w..."

**I went to see the movie and thought it was...**

- SFT: "a Friday night and I got there early, so I was able to get a great seat. The movie was pl..."

- ORPO (18.7%): "great and I enjoyed the movie! I'm sorry that the movie left me with mixed emotions. It's und..."

- COALA (66.4%): "so good I saw it twice in the theatre. I know there are a lot..."

- DPO (14.0%): "good! I loved it! My dad is a big fan of the old movie and the s..."

**After watching the film, I felt...**

- SFT (21.5%): "it's still one of my all time favorite movies. The movie was so unbelievable that you were..."

- ORPO (9.3%): "the movie is great! I love the way the movie plays with kids. It's interesting to see how the..."

- COALA (38.3%): "that it was very well done. I felt that the acting was very good...."

- DPO (30.8%): "a real sense of satisfaction. The film is, for me, a classic. The..."

**The director's vision for this movie...**

- SFT (8.4%): "is a good way to start a review, and I was not disappointed with this film. I was expecting so m..."

- ORPO (57.0%): "was to create a thought-provoking and emotionally engaging narrative that explor..."

- COALA (34.6%): "is very clear and it is well executed, the movie is very b..."

- DPO: "is something that I've always wanted to see for a long time,..."

**The actors in this movie performed...**

- SFT (10.3%): "gritty, hard-hitting, and unflinchingly honest looks at the world. The film follows the live..."

- ORPO: "great! I can't even begin to express how awesome they were. The actors are all very f..."

- COALA (43.9%): "admirably and believably, the characters were well-rounded..."

- DPO (40.2%): "to perfection. In this movie, the characters are really the stars..."

**The cinematography in this film was...**

- SFT (18.7%): "amazing. The film starts out with a woman in a room, and as the camera pans out, we see..."

- ORPO (26.2%): "pretty thrilling, about a group of friends who are trapped in a haunted house and must find..."

- COALA (29.9%): "beautiful. It had a great feel to it, and was very realistic, which..."

- DPO (25.2%): "incredible. It was shot like a documentary, but with a lot of..."

**My favorite scene in the movie was when...**

- SFT: "Hicks and Bishop are trying to kill the alien in the corridor, while a woman is screaming ¿.."

- ORPO (15.0%): "through the use of stunning visuals, powerful performance..."

- COALA (30.8%): "Alice is walking through the forest looking for Dave and sh..."

- DPO (28.0%): "the main character, played by the fabulous and talented Jud..."

### G.3. Human Evaluation Consent Form

**Title of Study:** Human Evaluation of Preference-Aligned Language Model Show Down

You are being asked to participate in a research study that evaluates human preferences for different fine-tuning algorithms used to align large language models (LLMs), such as COALA, DPO, ORPO, SimPO, CPO, GRPO and SFT. Your responses will help assess the quality and alignment characteristics of LLM-generated outputs.

**What Participation Involves:**

- You will complete a survey consisting of multiple-choice questions based on outputs from various LLM alignment methods.

- You will be asked to select responses based on perceived **correctness**, **helpfulness**, and **humanness**.

**Use of Data and Publication:**
By signing this form, you agree to the following:

- Your responses may be used in published research, academic papers, and open-access benchmarks evaluating LLM alignment.

- The results may be shared online, including datasets and analysis.

- Your name and institutional affiliation will be publicly disclosed as part of the human evaluation dataset, for transparency and attribution.

**Confidentiality & Voluntary Participation:**
Participation is voluntary. You may withdraw at any time before submission. After submission, your data—including your identity—may be published as part of a public benchmark and cannot be withdrawn.

**Acknowledgment & Consent:**
By signing below, you confirm that:

- You have read and understood the purpose and procedures of this study.

- You understand that your responses, name, and affiliation may be publicly shared as part of the dataset and may appear in future publications.

- You consent to participate in this research under these terms.

Participant Name: ___________________________________

Institutional Affiliation: ___________________________________

Email (optional): ___________________________________

Signature: ___________________________________

Date: ___________________________________

## H. Experimental Setup Details

### H.1. Hardware Configurations

COALA is trained using a single RTX-4090 GPU (24GB) with JAX, while SFT, DPO, and ORPO use individual A100 GPUs (40GB) with PyTorch. To ensure fair comparison, each (Method × Dataset × Model) setup is run at least three times. Unlike other methods, COALA requires no PEFT tuning, and instead leverages CRONOS (Section 4) for efficient optimization over high-dimensional features for its stage one training. Table 12 explicitly details hardware setup across experiments, and the following subsection H.2 details PEFT configurations used for the strongest competitors.

| METHOD | GPU | VRAM | ACCELERATION METHOD |
|---|---|---|---|
| COALA | RTX 4090 | 24GB | JAX WITH XLA |
| DPO | A100 | 40GB | DEEPSPEED + LORA |
| SFT | A100 | 40GB | DEEPSPEED + LORA |
| ORPO | A100 | 40GB | DEEPSPEED + LORA |

*Table 12.* Hardware and framework settings used across experiments.

### H.2. LoRA and DeepSpeed Configurations for Competing Methods

Table 14 gives the precise DeepSpeed configurations used to achieve the most competitive results via SFT, DPO, and ORPO methods.

| PARAMETER | VALUE |
|---|---|
| R | 16 (UP TO 256 IN DPO SETTING) |
| LORA_ALPHA | 32 (UP TO 128 IN DPO SETTING) |
| LORA_DROPOUT | 0.05 |
| BIAS | NONE |
| TASK_TYPE | CAUSAL_LM |
| TARGET_MODULES (GPT-2 BASE) | [C_ATTN, C_PROJ] |
| TARGET_MODULES (ALL OTHER MODELS) | [Q_PROJ, K_PROJ, V_PROJ, O_PROJ, GATE_PROJ, UP_PROJ, DOWN_PROJ] |

*Table 13.* LoRA Configuration Settings for Competing Methods.

### H.3. Inference Details

Our guided sampling pipeline integrates the convex model directly into generation. At each step, we use prefix-based nucleus sampling (Holtzman et al., 2019) (`top-p`=0.9, `top-k`=50, `num_candidates`=5) to form a nucleus-restricted pool of next-token candidates.

to generate candidates, score them using a contrastive-style objective and select the highest-scoring continuation. While both prefix-based and token-by-token modes are supported in our inference module, COALA experiments specifically utilized prefix-based. We emphasize that COALA's convex program is integrated into this structured guided sampling framework rather than beam search. Although COALA targets single-GPU resource constraints, we note the modular JAX inference pipeline is written to easily scale to cluster GPU environments, and supports low-memory devices via device-aware model loading. Since COALA's main contribution is its convex-based training algorithm for preference alignment, in order to be consistent with prior work such as DPO and SimPO, we focus primarily on the training stage in the main body of this work. Our benchmarks and ablations target training effectiveness and compute efficiency.

We note that there is substantial room for future work on improved inference-time generation strategies: including various batched guided decoding methods (Berdoz et al., 2025), utilizing a continuous guidance scale to shift the raw logits of the LLM toward preferred features at every token step (similar to how KL-penalties are applied in standard RLHF), as well as multi-step lookahead scoring and beam search integration with the convex head. Further inference-time generation strategies are promising directions (Liu et al., 2024; Zhu et al., 2024), which we look forward to exploring in future work.

**Guided Inference**

- **Prefix-conditioned candidate proposal:** At decoding step $t$, given the current prefix $p_t$, the frozen base LM performs standard autoregressive decoding and forms a nucleus-restricted pool of *next-token* candidates (top-$p = 0.9$, top-$k = 50$).

- **Convex-guided reweighting:** For each candidate $c$, we form the one-step extension $p_t \oplus c$ and score it with the fixed cvxNN head $g^{\text{cvx}}_{\Theta_1, \theta_2}$ using the last-layer hidden state of the extended prefix. These scalar scores are min–max normalized and used to reweight the base LM's candidate probabilities with guidance scale $\lambda$ (applied every $N = 5$ tokens).

- **Sampling and continuation:** We sample one token from the reweighted distribution, append it to the prefix, and repeat this procedure autoregressively until the stopping criterion is reached.

| Parameter | Value |
|---|---|
| *BF16* | |
| bf16.enabled | true |
| *Optimizer (AdamW)* | |
| optimizer.lr | $2.0 \times 10^{-4}$ |
| optimizer.betas | [0.9, 0.999] |
| optimizer.eps | $1 \times 10^{-8}$ |
| optimizer.weight_decay | 0.01 |
| *Scheduler (WarmupDecayLR)* | |
| scheduler.warmup_min_lr | 0 |
| scheduler.warmup_max_lr | $2.0 \times 10^{-4}$ |
| scheduler.warmup_num_steps | 1 000 |
| scheduler.total_num_steps | auto |
| *Zero Optimization (Stage 3)* | |
| zero_optimization.stage | 3 |
| zero_optimization.offload_optimizer.device | none |
| zero_optimization.offload_optimizer.pin_memory | true |
| zero_optimization.offload_param.device | none |
| zero_optimization.offload_param.pin_memory | true |
| zero_optimization.overlap_comm | true |
| zero_optimization.contiguous_gradients | true |
| zero_optimization.sub_group_size | $1 \times 10^{9}$ |
| zero_optimization.reduce_bucket_size | $3 \times 10^{8}$ |
| zero_optimization.stage3_prefetch_bucket_size | $3 \times 10^{8}$ |
| zero_optimization.stage3_param_persistence_threshold | $3 \times 10^{8}$ |
| zero_optimization.stage3_max_live_parameters | $1 \times 10^{9}$ |
| zero_optimization.stage3_max_reuse_distance | $1 \times 10^{9}$ |
| zero_optimization.stage3_gather_16bit_weights_on_model_save | true |
| gradient_accumulation_steps | 8 |
| gradient_clipping | 1.0 |
| train_micro_batch_size_per_gpu | 2 |
| train_batch_size | 16 |
| wall_clock_breakdown | true |

*Table 14.* DeepSpeed configuration used for training competing methods.

