# OpenReview forum: "Convex Optimization for Alignment and Preference Learning on a Single GPU"
_ICML.cc/2026/Conference — ICML 2026 regular_

### Official Review · Reviewer_5bQb · 2026-03-08

**Soundness:** 3
**Presentation:** 4
**Significance:** 3
**Originality:** 3
**Overall Recommendation:** 5
**Confidence:** 4

**Summary:**

Your paper proposes COALA , a new framework for preference alignment that is derived from a smart reformulation of the preference objective to a convex formulation. The key idea of your work is to attach a convex neural network (cvxNN) head on top of frozen LLM representations and optimize it using convex methods such as ADMM (via the new CRONOS algorithm). Your results are promising in reducing overall compute while increasing performance.

**Compliance With Llm Reviewing Policy:**

Affirmed.

**Final Justification:**

**The authors have contributed a solid piece of work and I recommend clear acceptance**.

The rebuttal has resolved all of my remaining concerns and in general provide clear evidence that their method works well and is generalizable. Also, the authors have commited to release some new experiments for the final version, which should be enough to make the case that this is solid work.

At the time of my final justification, other reviewers have not yet replied to the latest comments of the authors during the rebuttal. **I am thus willing to champion this work if need be.**

**Key Questions For Authors:**

Please see questions in Strengths and Weaknesses.

**Limitations:**

yes

**Strengths And Weaknesses:**

Thank you for your submission! Overall, I have understood the concept very well and have the following remarks:

### **1. Strengths**:
- **interesting new theoretical perspective on alignment**: your convex reformulation trick is very smart and has a lot of potential for other future work. Specifically, COALA reduces the alignment problem to logistic regression on top of convex features, which yields global optimality.
- **strong theoretical guarantees**: your formal convergence analysis is coherent and supports your claim that COALA avoids the instability of typical RLHF or DPO training.
- **efficiency improvements**: training efficiency is notable. Unlike DPO, COALA does not require a reference model, which reduces memory requirements. This is an important practical contribution because alignment methods often require expensive hardware.

There are more results in the appendix which further support your statements.

### **2. Weaknesses**:

Soft weaknesses are *italic*, while strong weaknesses are **bold**:

- *novelty*: I understand that the convex formulation is theoretically interesting, but the practical training procedure effectively reduces to freezing the base model and training a small classification head. This is conceptually similar to several existing approaches such as reward modeling, linear probing, preference classification heads. However, your empirical results and training efficiency clearly outweigh this weakness, hence why it's a soft one.
- *alignment capacity may be limited*: As acknowledged in Section 6.6, this approach may struggle to capture deeper semantic changes or stylistic alignment behaviors. COALA trains only a small convex head, while DPO and ORPO update many model parameters. Therefore, the large compute savings may partially result from reduced model capacity rather than improved optimization. However, I do not think that this is critical if the results are good. But (as mentioned later), you need to back this up with more evals.
- **models are somewhat outdated and could be more diverse**: the models you evaluate on are old compared to modern LLM architectures. I understand that compute is scarce, but you have an A100 GPU that can run a modern Llama-3.1 variant. Testing on more recent models would strengthen the empirical claims. Also, your models are roughly the same size. You should vary between model sizes such as 3B or 0.5B variants to solidify your results and claims.
- **alignment datasets could be broader**: I think UltraFeedback is a very good preference alignment dataset, however, the other two datasets are somewhat unconventional. Particularly, EduFeedback is synthetically generated using GPT-4o and might introduce bias and may thus not reflect real user preferences. Also, IMDB is too basic and I am not sure whether this is enough of a dataset for preference alignment. I would suggest trying out a more generic one. The reference in [1] open-sources fully annotated preference datasets, some of which are very popular in the community. For example, UltraMix is a bigger one and very generic, but they also give smaller ones like HelpSteer. I think you should include those datasets in your study for broader generalization and reproducibility.
- **limited alignment benchmarks**: The experiments are relatively small-scale compared to typical alignment research. Although you report metrics on AlpacaEval, I would have expected more evals, e.g., MMLU and basic tasks such as maths. This would also help the concern above regarding reduced model capacity. It would be interesting to see these insights as well, especially since other alignment research reports those. Also, it is easy to run these, for example, you can use lm-eval-harness.

Your results are very good and seem to partially confirm your optimization. However, I would expect a more comprehensive evaluation with standard alignment datasets to more standard tasks. I understand that this might be more work, but I think it is required to show at least one more generalized alignment dataset with more evals.

### **3. One Important Question**:
- Did I understand correctly that in the paper, you explicitly state that all base models are first SFT-trained on the same datasets used for preference training?
- I believe you say: *"The five baseline models are SFT fine-tuned for one epoch on each of the three datasets before preference extraction."*
- Then you create preference triplets from those same datasets: EduFeedback-Alternate, UltraFeedback-Binarized, etc.
- If so, then this creates data leakage between stages. That means the model already saw the answers during SFT. **Preference learning then becomes partly re-fitting the same distribution.**
- This can artificially inflate all of your performance metrics and violates common alignment evaluation practice.
- In essence, it is not necessarily invalid, but it is methodologically weak. Combined with the synthetic dataset, it means the empirical results may be less convincing than they appear.

If this is the case, then my suggestion would be to take actual SFT checkpoints of open models. To this end, I am again referring to the overview paper in [1] that lists many such open models that are in reasonable ranges (0.5B, 3B, 7B, 8B). **You should evaluate your method on one of these open SFT checkpoints to avoid SFT training using the same data.**

### **4. Typos and Style**:
- In line 020 of the Abstract, I believe it should be expensive hyperparameter tuning, not expansive
- In line 075, Related Work, I believe it should be "heuristics-driven" not "drive"
- In line 113, Section 3.2, I believe it should be "no information is lost" instead of "in lost"
- Table 1 needs explanation on the abbreviations (WR; LC WR) - I know these are standard, but nevertheless
- Table 3 could benefit from giving percentages in reduction, rather than absolute values

### **5. A Question for Future Work**:

This would just be out of curiosity:
- It seems like your method could be generalizable to more advanced algorithms.
- Would it for example work for GRPO and similar reasoning adaptations?

[1] When Data is the Algorithm: A Systematic Study and Curation of Preference Optimization Datasets, 2025

---

> ### Author Rebuttal · Authors · 2026-03-30
>
> We sincerely thank the Reviewer for taking the time to thoughtfully review our work! We are happy you appreciate the novelty, efficiency, and strong theoretical guarantees of our paper. We strive to address your questions/concerns in detail below.
>
> 1. **More Modern Models and Additional Dataset:** Per your request, we have now increased experiments to include Llama3.2-3B. This is a more modern model as noted, and also diversifies the model sizes. We have also added HelpSteer dataset as suggested, and updated all results in tables in the manuscript to reflect these comprehensive **216 runs on single GPUs**. In summary, our updated datasets are now:
>
> | Dataset | Preference Data Pairs | Train (90%) | Eval (10%) |
> |---|---|---|---|
> | EduFeedback | 65,606 | 59,045 | 6,561 |
> | UltraFeedback | 60,917 | 54,825 | 6,092 |
> | IMDb | 25,000 | 22,500 | 2,500 |
> | HelpSteer | 7,708 | 6,937 | 771 |
>
> Our more comprehensive range of model sizes:
>
> | Model | Parameters | Feature |
> |---|---|---|
> | DistilGPT-2 | 82M | Mini |
> | GPT-2 | 124M | Seminal |
> | LLaMA-3.2-3B | 3B | Modern as suggested by the Reviewer|
> | Mistral-7B | 7B | Performant and Balanced |
> | Dolphin-7B | 7B | Instruction Fine-tuned |
> | LLaMA-3.1-8B | 8B | Largest |
>
> One example of new ArenaHard results on Llama3.2-3B:
>
> | METHOD | EDU | IMDB | ULTRA | HELPSTEER |
> |---|---|---|---|---|
> | **COALA** | 67.50 | 60.91 | 62.35 | 61.25 |
> | **ORPO** | 56.25 | 50.50 | 35.29 | 58.82 |
> | **DPO** | 67.78 | 60.00 | 57.89 | 60.16 |
> | **SFT** | 27.78| 21.25 | 25.56 | 56.25 |
>
> We have strived to meticulously and rigorously conduct all experiments with COALA on a single RTX-4090, and all other methods on single A100s.
>
> 2. **Metrics:** Thank you for appreciating our AlpacaEval metrics! We'd like to kindly note that our metrics also span ArenaHard (Table 7), MT-Bench (Table 6), TFLOPs Measurements (Table 3), and crucially a 107-sample real human feedback evaluation (Table 2). We have now updated all tables to reflect these additional experiments, with the exception of Table 2. In Section 6.6 we discuss Expressiveness Tradeoff, and note that while the integration of the convex neural network provides global convergence guarantees, it is not intended to capture complex semantic shifts. Therefore we believe metrics such as deep reasoning and mathematical problems are beyond the scope of this work. We currently aim to take a step towards more interpretable, theoretically-backed methods in preference alignment (as opposed to largely heuristics and compute driven strategies). Which leads us to...
>
> 3. **Question for Future Work:** We are thrilled you have astutely noted the potential to generalize COALA into more advanced methods! Indeed, in this work we focus on the single GPU resource constrained setting for on-prem deployment and increased accessibility. The 107-sample real human feedback study contains current users of COALA generalized in the wild! This work is ideally suited for personal assistant or tutor scenarios, where speed, efficiency, data privacy and short conversations are key. We note that recent work identified over 77% percentage of LLM queries are actually simple tasks, which further motivates the potential impact of our methodology (references to follow for space considerations).
>
> 4. **Typos/Style:** Thank you for carefully pointing this out! We have corrected these points on lines 020, 075, 113. We have added captions in Table 1 for the abbreviations, and a % in reduction metric in Table 3.
>
> 5. **SFT-Trained Baselines** You are right that the current presentation conflates two settings. Some exp use same-domain SFT before preference optimization; this was intended as a controlled initialization study, not as a claim that standard preference alignment should always reuse the same data. We also evaluate non-SFT init; we now make this distinction explicit and report on them separately in the revision. Importantly, the final evaluations (AlpacaEval2, MT-Bench, ArenaHard) are performed on held-out prompts, so the concern relates more to init protocol rather than evaluation leakage.
>
> Finally, we'd like to summarize our main contributions briefly:
> - To the best of our knowledge, COALA represents the first time convex NN/ADMM techniques have been practically applied to preference alignment of LLMs.
> - Our theoretical guarantees further provide interpretability in preference fine-tuning, instead of being generally heuristics driven.
> - We introduce the EduFeedback Dataset, as well as the novel Alternating Population Strategy for efficient preference dataset generation from natural conversational settings.
> - Comprehensive exps include 6 models x 4 datasets x 5 methods, with a 107-sample real world human exp to validate efficacy.
> - This timely line of work aims to reduce dependence on GPU compute, and increase accessibility for lab practitioners with on-prem deployment options.
>
> Thank you again for reading our work, and we greatly look forward to discussion period!

---

> > ### Author Rebuttal · Reviewer_5bQb · 2026-03-31
> >
> > Thank you for the comprehensive addition of the rebuttal!
> >
> > I believe most of my questions have been resolved, especially with the new results on the HelpSteer dataset. **I will consider raising my score accordingly!**
> >
> > *I still think that the paper would benefit from additional results on the UltraMix dataset*, which is larger than the other datasets you examined. This would really further solidify your results and eliminate any doubt beyond the rebuttal and would be **especially useful for practitioners** who are interested in whether your method scales with dataset size.
> >
> > If the time permits, I would be more than happy to see these results!
> >
> > Thank you!

---

> > > ### Author Response · Authors · 2026-04-08
> > >
> > > Thank you for your thoughtful and valuable engagement throughout our discussion, which has greatly strengthened this paper! We are delighted that the additional results on the new HelpSteer dataset and Llama-3.2 model have addressed your primary concerns.
> > >
> > > We agree that additional experiments on the larger UltraMix dataset would further solidify our results, and have been actively running these experiments. We rigorously attempted to include these results within the discussion period, however since a core feature of our work is its single-GPU setting, we were unable to complete the full training and evaluation cycles for this significantly larger dataset before the deadline.
> > >
> > > We are fully committed to including the UltraMix results in the final Revision, and are excited to provide a more comprehensive span of experiments that further showcase the robustness of our method across dataset scales.
> > >
> > > Finally, we greatly enjoyed this rewarding discussion period and reading the comprehensive data-centric analysis of [1] (which has already meaningfully extended our Related Work section). We sincerely hope to have the opportunity to continue this interesting discussion in the future!
> > >
> > > **References:**
> > >
> > > [1] Djuhera, Aladin, et al. "When Data is the Algorithm: A Systematic Study and Curation of Preference Optimization Datasets." arXiv preprint arXiv:2511.10985 (2025).

---

### Official Review · Reviewer_MRNe · 2026-03-11

**Soundness:** 3
**Presentation:** 3
**Significance:** 2
**Originality:** 3
**Overall Recommendation:** 4
**Confidence:** 3

**Summary:**

In this work, the authors propose a method for preference learning that reformulates the problem as a convex optimisation problem. Namely, they introduce a cvxNN on top of a policy model and remove the reference model to train this new variant of direct preference optimisation. This approach allows them to train LLMs (up to 8B) parameters on a single GPU. The results that the authors present across 5 models, 3 datasets, and 5 benchmarks look promising at first sight, demonstrating competitive performance to traditional e.g DPO and significantly less FLOPS.

**Compliance With Llm Reviewing Policy:**

Affirmed.

**Final Justification:**

My points were addressed, and I increased my score.

**Key Questions For Authors:**

**Q1**: Against what baselines are the winrates calculated?

I am a bit confused about the win-rates: from my understanding, reading the appendix, an LLM as a judge compares all the 4 options against each other and decides a winner. Shouldn't the win-rate sum to 1 in e.g. Table 1 across the baselines?

**Limitations:**

I was not able to find any limitations section in the paper, and I would encourage the authors to discuss it further.

**Strengths And Weaknesses:**

# Strengths:
**S1:** The paper is well written and motivated. The methodological section is well explained, and to the best of my knowledge, makes sense.

**S2:** There is a large number of experiments on different baselines and models, as well as datasets, in order to demonstrate a lot of evidence.

# Weaknesses:
While I believe this is a good start, I am not yet fully convinced of the method's effectiveness. I have two main concerns I wish to point out:

**W1:** No error bars or any kind of statistical significance were reported in any of the experiments. While I understand that running experiments multiple times with different seeds might not be feasible, I would encourage you to at least report, e.g. the standard error on the test set, or do a bootstrapped 95% confidence interval. Such measures would indicate the stability of the results much better. This could, in my opinion, increase the quality of the paper.

**W2:** I am not convinced that the baselines are a fair comparison. From what I read in the appendix, all baselines are trained with LoRA, whereas the proposed method freezes the entire backbone of the LLM. I therefore feel like the Flop comparison is not fair. I would wonder how the Flop comparison is, if only the LM-head of the baselines was trained. Moreover, one of the claims in the introduction is that this approach fits on a GPU, as no reference model is needed. However, from my understanding, not using the reference model is not really a discovery of this methodology, but rather something already observed in prior work. Therefore, a fair baseline would be DPO, where only the LM head is trained, and no reference model is used for normalisation (this would surely significantly reduce flops, I assume?).
What might be interesting for readers is that, rather than having win-rates and flops in two tables, it could be useful to show a scatter plot with win-rate on one axis and flops on the other. This could give a better indication of how well the model is performing, rather than how much computation it needs.

---

> ### Author Rebuttal · Authors · 2026-03-30
>
> Thank you for your helpful feedback, which has greatly strengthened this submission! We are grateful for your kind words highlighting the breadth of our experiments and clearly motivated methodology. We will strive to address your valid concerns on stronger reporting of uncertainty/comparison scope below.
>
> 1. **Win-rate Details:** We evaluate via Custom Pairwise Win Rate Excluding Ties with standard GPT-4 as base reference model and GPT-4o as judge via the provided `--reference flag`. Therefore metrics provide the percentage of each method's "wins" over the reference model, and not against each other. The Reviewer is completely correct that we should discuss this, and we have now revised the manuscript clearly note that: "these numbers are not pairwise shares among the four methods, so they do not sum to 1. In AlpacaEval/ArenaHard our WR columns report win rate against the fixed reference/judge setup, not against the other three methods jointly." We have made these details explicit in the Table captions. In addition we have clarified all Table headings (such as LC WR% and WR%) to be fully transparent on results.
>
> 2. **On Limitations Visibility:** We'd like to kindly note, that limitations are discussed in Section 6.6. We will adjust the heading of this Section to explicitly state "Limitations" in bold, so as to improve readability of the manuscript. We have now also expanded the limitations section to discuss the best use-cases of our novel Alternating Population Strategy for efficiently generating preference datasets.
>
> 3. **LoRA Training:** We agree that COALA and the baselines do not update the same number of parameters. Our intended claim is therefore not the same parameter budget, but a practical single GPU option for targeted use-cases (Section 6.6) against strong preference optimization baselines in their feasible configurations. For the larger 7B/8B models, LoRA is the realistic single-GPU configuration for competing methods.
>
> This resource-constrained setting provides on-prem deployment options, and aims to increase accessibility for labs/individuals. TFLOPs measurements are carefully recorded, and per your request we have generated the scatter plot for better visibility (https://postimg.cc/NyZj6gm6).
>
> 4. **Reference Model-free:** We definitely do not intend to claim to be the first reference-free method! We select competing methods (ORPO, the newly included SimPO, and DPO) since they can be implemented as reference-free. Specifically, we  experiment with the reference-free version of DPO since otherwise the memory requirements of hosting a second copy of the model (to train one single model) will exceed the single GPU requirements. We have now expanded the current discussion of this from Appendix E, into a comprehensive paragraph in Section 2 and revised the introduction/related work to avoid overstating this point.
>
> Relatedly, the novelty of COALA is the convex preference head and its theory-backed optimization properties, not merely removing the reference model.
>
> 5. **TFLOPs Efficiency:** COALA's TFLOPs efficiency is not reliant on its reference-free nature, but instead on the novel integration of the convex reformulation neural network. The convex module ensures global optimality, which provides preference alignment performance, and also permits an ADMM solve in JAX. Since this is not a gradient-descent update, it becomes extremely robust and efficient and is practically hyperparameter-free (Table 8). The limitations is that there necessarily exhibits expressiveness-tradeoffs, which we discuss in Section 6.6. We have now also added Appendix H for further discussion on convex neural networks and ADMM strategies.
>
> 6. **Variance and Error bars:** This is a very fair criticism.  Our (method × dataset × model) results are averaged over 3 runs; we should have reported the corresponding variability explicitly. We have reported mean±std (and 95% CIs where appropriate) to the main tables/plots so the stability claims are quantitatively visible rather than only qualitative.
>
> Finally, we'd like to summarize our main contributions briefly:
> - To the best of our knowledge, COALA represents the first time convex NN/ADMM techniques have been practically applied to preference alignment of LLMs.
> - Our theoretical guarantees further provide interpretability in preference fine-tuning, instead of being generally heuristics driven, and offer convergence guarantees.
> - We introduce the EduFeedback Dataset, as well as the novel Alternating Population Strategy for efficient preference dataset generation from natural conversational settings.
> - Comprehensive experiments span 6 models x 4 datasets x 5 methods as well as a 107-sample real world human experiment to validate efficacy.
> - This timely line of work aims to reduce dependence on GPU compute, and increase accessibility for lab practitioners with on-prem deployment options.
>
> Thank you again for reading our work and we greatly look forward to discussion!

---

> > ### Author Rebuttal · Reviewer_MRNe · 2026-04-01
> >
> > Thank you very much for your responses. I appreciate the detail you put into them, which helps me clarify some concerns.
> > I will address them individually here:
> >
> > 1. **Win-Rate Details**: That makes sense, thank you for clarifying. I see this as resolved.
> > 2. **On Limitations of Visibility**: Thank you for pointing this out. I apologise for not having noticed it. Limitations have therefore also been addressed.
> > 3. **LoRA Training**: I see your point that you are trying to make here, and I agree that for a single GPU for 7B/8B models, LoRA is an adequate method to train an LLM, if one wants to reach the best performance. However, I still find the comparison of your method against LoRA in terms of TFLOPs slightly misleading, as LoRA is optimised for performance rather than compute. However, I think the provided image makes a strong case for your paper, as it combines performance and TFLOPs. Thank you for providing it! I see this matter as addressed as well.
> > 4. **Reference Model-Free**: Thank you for clarifying this. It seems I misunderstood the baselines; they were all in their original form, including the reference model. If this is not the case, then this matter has also been addressed.
> > 5. **TFLOPs Efficiency**: Thank you for clarifying, and sorry for misunderstanding it.
> >
> > Open question:
> >
> > 6. **Errorbars**: This is great! While I trust your word, I would love to see the tables with error bars to see how large the variance is. Any chance of providing this? I'd be happy to raise my score then, if they look convincing.

---

> > > ### Author Response · Authors · 2026-04-05
> > >
> > > Thank you for carefully reviewing our rebuttal! Absolutely, we appreciate the opportunity to address your open question, and we are happy to present the revised table below. We now report mean ± std over the same three independent runs used throughout the paper, which aims to make variability explicit rather than only qualitative.
> > >
> > > **Table 1:** AlpacaEval2 metrics (mean $\pm$ std over 3 runs) by alignment method for SFT-initialized models across three datasets. For ease of readability, we also present this Table at (https://postimg.cc/6yv0jBHw)
> > >
> > > | | Method | |LC WR %  | | |WR %  | | |Avg Len  | |
> > > |---|--------|---------|---------|---------|------|------|------|------------|----------|----------|
> > > | | | Edu | IMDb | Ultra | Edu | IMDb | Ultra | Edu | IMDb | Ultra |
> > > | **Mistral7B** | | | | | | | | | | |
> > > | | **COALA** | 24.61±0.30 | 24.88±1.46 | 20.84±1.35 | 23.82±1.38 | 23.11±1.96 | 20.91±1.65 | 592±37 | 418±30 | 459±35 |
> > > | | ORPO | 17.01±2.82 | 17.58±2.32 | 16.04±2.30 | 14.67±2.87 | 15.62±2.48 | 12.28±2.34 | 561±30 | 368±39 | 350±31 |
> > > | | DPO | 24.19±1.19 | 24.30±1.26 | 17.68±1.67 | 22.45±1.50 | 22.74±1.26 | 15.56±2.11 | 492±37 | 502±55 | 453±34 |
> > > | | SFT | 6.80±0.51 | 8.42±1.16 | 6.18±1.31 | 10.30±1.03 | 1.77±1.73 | 6.11±1.59 | 515±20 | 428±31 | 463±34 |
> > > | **Dolphin7B** | | | | | | | | | | |
> > > | | **COALA** | 40.81±0.22 | 39.72±0.36 | 31.58±0.34 | 39.05±0.29 | 38.46±0.41 | 30.21±0.38 | 439±38 | 454±33 | 445±37 |
> > > | | ORPO | 25.06±1.93 | 24.90±1.23 | 22.94±1.26 | 23.59±1.18 | 22.92±1.77 | 25.93±2.51 | 526±9 | 448±37 | 452±45 |
> > > | | DPO | 34.73±0.80 | 33.86±0.72 | 26.41±0.68 | 32.46±1.09 | 31.58±1.52 | 24.86±1.02 | 494±34 | 511±49 | 476±35 |
> > > | | SFT | 17.36±0.38 | 16.21±0.23 | 14.88±0.46 | 15.30±0.10 | 15.47±0.44 | 12.76±1.08 | 404±39 | 423±25 | 459±24 |
> > > | **LLaMA8B** | | | | | | | | | | |
> > > | | **COALA** | 40.90±0.09 | 27.64±0.27 | 20.64±0.08 | 38.20±0.11 | 25.68±0.36 | 18.32±0.23 | 562±12 | 415±33 | 552±20 |
> > > | | ORPO | 23.87±0.60 | 12.10±0.71 | 12.91±0.50 | 20.58±0.68 | 12.05±0.55 | 10.95±0.55 | 599±70 | 610±33 | 354±41 |
> > > | | DPO | 40.68±0.10 | 21.79±0.29 | 18.89±0.31 | 38.53±0.47 | 20.18±0.49 | 15.81±0.30 | 539±24 | 449±98 | 503±39 |
> > > | | SFT | 10.92±0.20 | 8.16±0.11 | 7.41±0.30 | 10.75±0.39 | 8.11±0.66 | 5.62±0.78 | 384±55 | 435±49 | 546±15 |
> > >
> > >
> > > **Statistical Stability:** Table 1 demonstrates COALA's substantially lower variance across the majority of settings (models x datasets) in comparison to competing methods. For example, on the LLaMA-8B (Edu) setting, COALA achieves a tight $40.90 \pm 0.09$ in LC WR\%. This is also consistent with Theorem 4.3: that COALA loss achieves global optimality in polynomial time, while providing mathematical interpretation that the convex optimization framework yields a more robust alignment trajectory than heuristic-driven non-convex methods. These results are also supported in the recent works of [1,2,3]. Indeed, the deterministic nature of convex methods is a huge motivating factor of this work! Interestingly, Length Controlled Win Rates display lower variance than pure Win Rates. We intuitively attribute this to the high-variance length exploitation behavior often observed in preference optimization.
> > >
> > > We have added these updates in the Revision, which has improved the rigor and clarity of our work, and hope this addresses your open question!
> > >
> > > [1] Ergen et al., "Global optimality beyond two layers: Training deep relu networks via convex programs." International Conference on Machine Learning. PMLR (2021).
> > >
> > > [2] Gautier et al., "Globally optimal training of generalized polynomial neural networks with nonlinear spectral methods." Advances in Neural Information Processing Systems 29 (2016).
> > >
> > > [3] Feng et al., "Cronos: Enhancing deep learning with scalable gpu accelerated convex neural networks." Advances in Neural Information Processing Systems 37 (2024).

---

### Official Review · Reviewer_LsHe · 2026-03-12

**Soundness:** 3
**Presentation:** 3
**Significance:** 3
**Originality:** 3
**Overall Recommendation:** 4
**Confidence:** 3

**Summary:**

The paper introduces COALA, a novel framework for preference fine-tuning of Large Language Models (LLMs) on a single GPU. COALA leverages convex optimization to reformulate the preference learning problem, eliminating the need for a reference model and reducing both training time and VRAM consumption. The authors claim that COALA achieves competitive performance and efficiency, using as little as 17.6% of the total TFLOPS required by Direct Preference Optimization (DPO). The paper includes theoretical guarantees for convergence and empirical results across three datasets and five models, including LLaMA-8B. The authors also introduce a new dataset, EduFeedback, and a method for generating preference training pairs, the "Alternating Population Strategy."

**Compliance With Llm Reviewing Policy:**

Affirmed.

**Final Justification:**

The points raised in limitation part are well addressed.

**Key Questions For Authors:**

- **Clarification on Convex Reformulation**: The authors should provide more details on the potential suboptimal solutions that may arise from the convex reformulation. Specifically, they should discuss the conditions under which the convex reformulation may not yield the optimal solution and the impact of these suboptimal solutions on the performance of COALA.
- **Generalization to Other Models and Datasets**: The authors should conduct additional experiments on a wider range of models and datasets to further validate the generalizability of COALA. This would provide a more comprehensive understanding of the robustness and practicality of the approach.

**Limitations:**

- **Convex Reformulation**: While the convex reformulation provides strong theoretical guarantees and computational efficiency, it may not always yield the optimal solution. The authors acknowledge this limitation but do not provide a detailed analysis of the potential suboptimal solutions and their impact on performance.
- **Dataset and Model Generalization**: The experiments are conducted on a limited set of models and datasets. While the results are promising, it is not clear how well COALA generalizes to other models and datasets. The authors could benefit from additional experiments on a wider range of models and datasets to further validate the robustness of the approach.

**Strengths And Weaknesses:**

### 1. Soundness
- **Technical Soundness**: The paper presents a well-structured and theoretically grounded approach. The use of convex optimization and the Alternating Direction Method of Multipliers (ADMM) is well-justified and supported by existing literature. The theoretical analysis, including the convex reformulation and convergence guarantees, is rigorous and well-explained.
- **Empirical Results**: The experiments are well-designed and cover a range of models and datasets. The results are presented clearly, and the authors provide a detailed comparison with existing methods, such as DPO. The use of a single RTX-4090 GPU for COALA experiments and an A100 GPU for other methods ensures a fair comparison.
- **Strengths and Weaknesses**: The authors are careful to discuss both the strengths and weaknesses of their approach. They highlight the computational and memory efficiency of COALA but also acknowledge the limitations of the convex reformulation, such as the potential for suboptimal solutions in certain cases.

### 2. Presentation
- **Clarity and Structure**: The paper is well-written and well-structured. The introduction clearly outlines the problem and the motivation for the proposed solution. The related work section provides a comprehensive overview of existing methods and their limitations. The technical sections are detailed and well-organized, with clear explanations of the convex reformulation and the COALA algorithm.
- **Reproducibility**: The authors provide a modular open-source JAX codebase, which ensures ease of reproducibility. The detailed description of the datasets and the experimental setup, including the use of the EduFeedback dataset and the Alternating Population Strategy, further enhances the reproducibility of the work.
- **Context and Positioning**: The paper positions itself well in the context of existing literature. The authors clearly discuss how COALA differs from and improves upon existing methods. The introduction of the EduFeedback dataset and the Alternating Population Strategy adds to the novelty and practicality of the work.

### 3. Significance
- **Problem Relevance**: The paper addresses an important and relevant problem in the field of LLMs: the computational and memory efficiency of preference fine-tuning. The need for more efficient and scalable methods is well-justified, especially as LLMs become more widely adopted and the demand for personalized and aligned models increases.
- **Advancement**: COALA offers a significant advancement in the field by providing a theoretically grounded and computationally efficient method for preference fine-tuning. The reduction in training time and VRAM consumption, as well as the elimination of the need for a reference model, are substantial contributions.
- **Impact**: The work has the potential to influence future research and applications. The modular and open-source nature of the codebase, as well as the introduction of the EduFeedback dataset, make it easier for other researchers to build upon and extend the work. The practicality of COALA for single-GPU use cases is particularly valuable for on-premises and privacy-sensitive applications.

### 4. Originality
- **New Insights and Contributions**: The paper provides new insights into the use of convex optimization for preference fine-tuning of LLMs. The theoretical guarantees and the empirical results demonstrate the effectiveness of the approach. The introduction of the EduFeedback dataset and the Alternating Population Strategy are novel and valuable contributions.
- **Novel Combination of Techniques**: COALA combines convex optimization, ADMM, and the Bradley-Terry model in a novel way to address the challenges of preference fine-tuning.
- **Distinction from Related Work**: The authors clearly distinguish COALA from related work, such as DPO and RLHF, and provide a detailed comparison of the methods. The theoretical and empirical results support the novelty and effectiveness of COALA.

---

> ### Author Rebuttal · Authors · 2026-03-30
>
> Thank you for your valuable feedback, which has greatly strengthened this submission! We are happy you appreciate the technical soundness, and meticulous running of individual experiments on single RTX-4090 and A100 GPUs for this resource-constrained setting.
>
> In order to address your concerns, we have now **increased experiments from the initial 105 runs to 216 individual runs**. Our work now includes both a wider range of models and datasets as per your insightful suggestions! Below we aim to address your questions point-by-point.
>
> 1. **Convex Reformulation and Suboptimal Solutions:** This is a subtle point, and we would like to distinguish two types of optimality in the context of COALA:
> - On the Exactness of the Reformulation: Following the framework of existing work (references to follow), the training of a two-layer ReLU network is equivalent to our convex program provided that the number of hidden neurons $m$ is sufficiently large (typically $m \ge d+1$). In COALA, we leverage the high-dimensional feature space of the frozen backbone (such as $d=4096$ for the Llama-8B model), which ensures that the convex program is an exact global optimizer in this setting.
> - The Suboptimality Trade-off: The suboptimal outcomes discussed in our work refers to the gap between training a convex head on frozen features versus full-parameter fine-tuning. While full-parameter tuning has potentially higher expressiveness, it lacks global convergence guarantees and often suffers from reward margin instability (Figure 1).
>
> 2. **Discussion on Conditions for Gap Widening:** We now discuss the following two conditions:
> - Feature Misalignment: When the pre-trained features $\phi(x,y)$ do not contain the necessary information to distinguish between specific preference pairs (such as highly nuanced/sensitive creative writing styles).
> - Dataset Complexity: In tasks requiring deep semantic shifts rather than stylistic steering, the frozen backbone acts as a bottleneck.
>
> 3. **Impact on Performance:** Empirically, as shown in Table 1 and Table 7, this trade-off is highly favorable for the objective and correctness-focused alignment tasks COALA targets. The suboptimality relative to an expensive full-model update design choice provides trade-offs/buys us the convergence guarantees to global optimality and demonstrated high TFLOPs efficiency required for sustainable on-prem deployment settings (Table 3).
>
> 4. **On Dataset and Model Generalization:** While our initial work encompassed 105 individuals runs, in order to provide more feedback on robustness and generalization we have now increased this to 216 individual experiments. We have added an additional preference alignment dataset (HelpSteer) for structured experiments across the following:
>
> | Dataset | Preference Data Pairs | Train (90%) | Eval (10%) |
> |---|---|---|---|
> | EduFeedback | 65,606 | 59,045 | 6,561 |
> | UltraFeedback | 60,917 | 54,825 | 6,092 |
> | IMDb | 25,000 | 22,500 | 2,500 |
> | HelpSteer | 7,708 | 6,937 | 771 |
>
> We have also added the Llama3.2-3B model, to examine effect of a varying range of model sizes:
>
> | Model | Parameters | Feature |
> |---|---|---|
> | DistilGPT-2 | 82M | Mini |
> | GPT-2 | 124M | Seminal |
> | LLaMA-3.2-3B | 3B | Modern as suggested by the Reviewer|
> | Mistral-7B | 7B | Performant and Balanced |
> | Dolphin-7B | 7B | Instruction Fine-tuned |
> | LLaMA-3.1-8B | 8B | Largest |
>
> We have additionally revised Section 6.6 (to "Limitations and Expressiveness Tradeoffs"), to provide detailed analysis of the feature-bottleneck condition and clarified the distinction between architectural optimality (which we guarantee) and model-wide expressiveness (which we trade for efficiency). Comprehensive experiments across **216 runs** validate the efficacy of our theoretically-backed methodology.
>
> Finally, we'd like to summarize our main contributions briefly:
> - To the best of our knowledge, COALA represents the first time convex NN/ADMM techniques have been practically applied to preference alignment of LLMs.
> - Our theoretical guarantees further provide interpretability in preference fine-tuning, instead of being generally heuristics driven, and offer convergence guarantees.
> - We introduce the EduFeedback Dataset, as well as the novel Alternating Population Strategy for efficient preference dataset generation from natural conversational settings.
> - Comprehensive experiments span 6 models x 4 datasets x 5 methods as well as a 107-sample real world human experiment to validate efficacy.
> - This timely line of work aims to reduce dependence on GPU compute, and increase accessibility for lab practitioners with on-prem deployment options.
>
> Thank you again for reviewing our work, and please let us know of your any other comments or suggestions!

---

> > ### Author Rebuttal · Reviewer_LsHe · 2026-04-02
> >
> > The points raised in limitation part are well addressed. Thanks.

---

> > > ### Author Response · Authors · 2026-04-06
> > >
> > > Thank you again for thoughtfully reviewing our work, we are happy that your concerns have been fully addressed! As this valuable discussion period nears its conclusion, we would like to briefly highlight our revisions and contributions:
> > >
> > > 1. **High Novelty:** COALA introduces a novel theoretically-backed approach to preference alignment. To our knowledge, this is the first practical application of convex reformulations and ADMM-style optimization to LLM preference learning: a contribution which advances the field towards more interpretable and deterministic strategies, in contrast to broadly heuristic-driven methods utilizing large scale compute.
> > >
> > >
> > > 2. **Empirical Performance Gains:** In response to your request for a wider range of Models and Datasets, we have conducted a comprehensive round of additional experiments including both the new Llama3.2-3B [1] model and Nvidia's HelpSteer[2] dataset. This expands our study **from 105 to 216 individual runs**, and demonstrates the generalization and efficiency of our work.
> > >
> > >
> > > 3. **Clarity and Rigor:** We have significantly improved the manuscript’s clarity and rigor by providing a deeper analysis of the feature-bottleneck condition in the updated Limitations section (Section 6.6), and adding a "Discussion on Conditions for Gap Widening" section in Appendix A.2.
> > >
> > > 4. **Community Impact:** By enabling competitive preference alignment on a single GPU (utilizing as little as 17.6% of DPO's TFLOPS), COALA democratizes alignment research for more practitioners. Especially as GPU and compute resources become increasingly challenging to scale, COALA offers a timely framework which presents novel paths for future research and practical methodology for real world scenarios.
> > >
> > >
> > >
> > > Since you have found our responses and the additional new experiments to be fully satisfactory, we would be extremely encouraged if you might consider raising your score to reflect this updated assessment. Thank you sincerely for your time, effort, and contribution in improving the quality of this paper substantially!
> > >
> > > **References:**
> > >
> > > [1] Dubey, Abhimanyu, et al. "The Llama 3 Herd of Models." (https://huggingface.co/meta-llama/Llama-3.2-3B), arXiv preprint arXiv:2407.21783, 2024.
> > >
> > > [2] Wang, Zhilin, et al. "Helpsteer: Multi-attribute helpfulness dataset for steerlm." (https://huggingface.co/datasets/nvidia/HelpSteer), Proceedings of the 2024 Conference of the North American Chapter of the Association for Computational Linguistics: Human Language Technologies (Volume 1: Long Papers). 2024.

---

### Official Review · Reviewer_1aXG · 2026-03-12

**Soundness:** 2
**Presentation:** 3
**Significance:** 3
**Originality:** 2
**Overall Recommendation:** 3
**Confidence:** 3

**Summary:**

This paper proposes COALA, a preference-learning framework that replaces standard end-to-end preference optimization with a convex neural network head on top of frozen pretrained LLM features. The method uses CRONOS/ADMM for training a convex surrogate model, then optimizes a convex logistic objective for preference alignment, with the goal of reducing memory and compute so that training can be done on a single GPU. The paper reports experiments on three datasets and multiple base models, highlighting lower compute usage, smoother reward-margin curves, and competitive benchmark and human-evaluation results relative to SFT, DPO, and ORPO.

**Compliance With Llm Reviewing Policy:**

Affirmed.

**Key Questions For Authors:**

- Could the authors clarify under what assumptions it is valid in Appendix A to replace $\pi(y_l|x)$ with $1-\pi(y_w|x)$?

- How exactly does COALA produce responses at inference time?

- Could you include the results of SimPO?

**Limitations:**

No. See "Strengths and Weaknesses" for suggestions.

**Strengths And Weaknesses:**

**Strengths**

- The paper has a clear and timely motivation. Reducing the cost of alignment and preference learning is a real problem, and the emphasis on single-GPU training is practically relevant for many labs and practitioners.

- I appreciate that the paper tries to connect a fairly different toolbox, convex reformulations and ADMM-style optimization, to the preference-learning setting. Even if I have concerns about the final formulation, the attempt is intellectually interesting and not just a routine tweak of DPO.

- The paper includes human evaluation rather than relying solely on automatic judges, as is commonly done.

**Weaknesses**

1. The convexity proof appears to address a different problem from the one claimed. In Proposition 4.1 and its proof in Appendix A, the pairwise chosen/rejected preference objective effectively reduces to a logistic regression formulation in ( $\theta_2$ ), where the rejected response disappears from the final expression. Such a reduction would only be natural if the chosen and rejected samples had complementary binary probabilities, which is not the case here. For general text responses, this step is not justified in the main paper.

2. The paper is not clear about whether it is actually fine-tuning a language model or training a preference scorer on frozen features. This is the biggest issue. The policy in Equation (8) is defined through a logistic function on top of frozen pretrained features and a convex head. That is much closer to a classifier over fixed embeddings than to standard LLM fine-tuning. If the practical method is “sample candidates from the frozen LM and score them with the convex head,” that is a valid direction, but it is not the same as DPO-style policy optimization. This matters because the scientific claim and the comparison target change substantially depending on which interpretation is correct.

3. SimPO is the obvious missing comparison, especially because the paper discusses Meng et al. (2024) and even includes SimPO in Table 8. This matters because some of the paper’s selling points, reference-free training and improved stability, are exactly where SimPO is relevant.

4. The custom preference-data construction is potentially confounded. Section 6.3 on Page 7 treats an earlier assistant's answer as chosen and a later answer as rejected because later answers are supposedly less direct. But later answers can respond to follow-up user turns. In that case, the rejected answer is simply answering a different question. This can create a dataset where “preference learning” partly reduces to “spot the mismatched response,” which is much easier than genuine fine-grained preference optimization.

---

> ### Author Rebuttal · Authors · 2026-03-30
>
> Thank you for carefully reading our paper and for your insightful comments! We appreciate your kind words regarding our novel approach, and the valuable positioning of this timely line of work. We will address each question/concern point by point below.
>
> 1. **Convexity:** We appreciate the chance to clarify explicitly the probability object COALA
> is modeling, as well as the overloaded notation. In Appendix A, the substitution is valid only for the Bernoulli Bradley–Terry pairwise comparison variable, and not for the autoregressive LM distribution over full text responses. Under this pairwise model, $p(y_w \succ y_l \mid x) = 1 - p(y_l \succ y_w \mid x)$ therefore the logistic form follows directly. We have revised Eqs. 7–9 and Appendix A to separate the pairwise preference probability from the base LM's sequence distribution and make this derivation and notation explicit.
>
> 2. **Methodology:** COALA does not perform gradient-based updates to the base LM weights, instead it optimizes a convex neural network (cvxNN) head on top of frozen pre-trained features. This design choice crucially provides global optimality guarantees and high efficiency. We have moved the Step-by-Step guide (formerly App. B) into Section 4 for improved accessibility, and updated the Abstract to explicitly clarify that COALA is a preference classifier integrated into a guided sampling framework.
>
> 3. **SimPO:** Per your request, we have completed an **additional 24 runs of SimPO across 4 datasets and 6 models** on a single A100 GPU. The results of the seminal ArenaHard metric on the newly added LLaMA3.2-3B model is presented below for your convenience:
>
> | METHOD | EDU | IMDB | ULTRA | HELPSTEER |
> |---|---|---|---|---|
> | **COALA** | 67.50 | 60.91 | 62.35 | 61.25 |
> | **ORPO** | 56.25 | 50.50 | 35.29 | 58.82 |
> | **DPO** | 67.78 | 60.00 | 57.89 | 60.16 |
> | **SFT** | 27.78| 21.25 | 25.56 | 56.25 |
> | **SimPO** | 10.00 | 0.00 | 17.65 | 0.00 |
>
> SimPO's performance was likely hindered by its high reliance on hyperparameter tuning, whereas COALA is robust to these heuristics (Table 8). These results are also in line with recent work [3]. We have added these additional experiments to the Appendix!
>
> 4. **Alternating Population:** This novel strategy arose from real world on‑prem deployment needs. While real human data was scarce and costly, synthetic‑data attempts diverged from real human preferences. We designed this sliding‑window scheme to extract multiple (prompt, chosen, rejected) triplets from short yet concise dialogs. This works best for dialogues with objectively correct answers (e.g., inventory checks/action confirmations) and is not intended for long subjective interactions. Results presented are not sensitive to this heuristic, as only the Edu‑Alternate uses this, and COALA remains competitive on other datasets. We'd like to kindly note that we discuss limitations/trade-offs of such methodology in Section 6.6, and have made the following revisions to address your concerns:
> - Added more examples of this method in Appendix E.3 (https://postimg.cc/gallery/rW969ry).
> - Increased limitations discussions of the Alternating Population Strategy and uses cases in Section 6.6.
> - Added a new dataset (HelpSteer) which validates that our performance is not reliant on one particular dataset.
> We'd also like to please note, that the results of the real human feedback experiment valid the usefulness of this method in the real world personal assistant setting.
>
> 5. **Inference:** Inference details are provided in Appendix G.3, which we have moved into the main paper section in the revision.  Specifically, the trained convex head scores candidate continuations produced by the frozen base LM (prefix-based nucleus sampling, top-p=0.9, top-k=50, described in Appendix G.3) and selects the highest-scoring continuation. The final generated text from this pipeline is what is evaluated by AlpacaEval2, MT-Bench, ArenaHard, and 107 human participants.
>
> To the best of our knowledge, COALA represents the first time cvxNN/ADMM techniques have been practically applied to preference alignment of LLMs. By reformulating the head as a convex program, we achieve global optimality in the final layer's weights (which standard non-convex DPO/linear-probing cannot provide). We hope the additional Dataset, Model, and Method experiments address your concerns!
>
> Key references:
>
> [1] Rafailov, Rafael, et al. "Direct preference optimization: Your language model is secretly a reward model." Advances in neural information processing systems 36 (2023).
>
> [2] Ouyang, Long, et al. "Training language models to follow instructions with human feedback." Advances in neural information processing systems 35 (2022).
>
> [3] Li, Xiaoyi. "Do Post-Training Algorithms Actually Differ? A Controlled Study Across Model Scales..." arXiv preprint arXiv:2603.19335 (2026).
>
> [4] Meng, Yu, et al. "Simpo: Simple preference opt..." https://github.com/princeton-nlp/SimPO

---

> > ### Author Rebuttal · Reviewer_1aXG · 2026-04-02
> >
> > Thank you to the authors for the detailed rebuttal and the additional experiments. I still have several questions regarding the inference procedure.
> >
> > 1. Although I checked the inference details in the appendix, I do not think the explanation is sufficiently clear. What exactly is meant by “prefix-based” nucleus sampling? Since it is introduced alongside the “token-by-token” mode, does this mean it is different from standard autoregressive generation, where one token is generated at each step?
> > 2. What does “candidates” refer to in this context? Are the candidates individual tokens, or are they spans of continuations?
> > 3. What is the “contrastive objective” here? How exactly is this objective defined and used during inference?

---

> > > ### Author Response · Authors · 2026-04-05
> > >
> > > Thank you for your continued interest and follow-up! We appreciate your detailed response, and will strive to address your questions regarding inference concisely below.
> > >
> > > 1. **On prefix-based nucleus sampling vs. token-by-token:**
> > > COALA uses standard autoregressive decoding. At decoding step $t$, given the current prefix $p_t​$, the frozen base LLM forms a nucleus-restricted pool of next-token candidates conditioned on that prefix (top-$p$=0.9, top-$k$=50). By “prefix-based,” we mean that each candidate is scored as the one-step extension of the current prefix. This is not beam search or non-autoregressive decoding. The “token-by-token” mode mentioned refers only to how the guidance module is applied in the implementation. In both cases, generation remains autoregressive.
> > >
> > > 2. **Candidates:**
> > > Candidates refers to individual one-step continuations of the current prefix, and not long spans or complete beams. The convex neural network score is computed on the full extended prefix, hence our utilization of the term "prefix-based": the scoring is conditioned on the prefix-extended sequence rather than on the token in isolation.
> > >
> > > 3. **Contrastive objective:**
> > > We agree this phrase was imprecise and have replaced it with "convex-guided reweighting score" in the Revision. There is no separate contrastive loss optimized at inference time. More precisely, we use the trained convex neural network head to reweight candidate probabilities. The convex head produces a preference score for each candidate which is then used to weigh the base LLM's probability for each candidate relatively. Therefore tokens whose extended prefixes score higher under the convex neural network receive a larger boost relative to lower-scoring candidates. We then sample from this reweighted distribution. This is "contrastive" only in the sense that scores are applied relatively across candidates. No reference model is involved and no additional contrastive training objective is applied at inference.
> > >
> > > **Summary:**
> > > Our primary contribution in this work is the novel integration of convex reformulation and ADMM-style optimization into the preference alignment pipeline. This approach aligns with recent studies which suggest that lightweight steering vectors and residual modifications can achieve comparable performance with efficiency (Section 6.6). Our method introduces a theory-backed convex module to the largely heuristic-driven landscape of alignment, and offers improved interpretability while establishing a principled foundation for future work.
> > >
> > > We note that there is substantial room for future work on improved inference-time generation strategies: including various batched guided decoding methods, utilizing a continuous guidance scale to shift the raw logits of the LLM toward preferred features at every token step (similar to how KL-penalties are applied in standard RLHF), as well as multi-step lookahead scoring and beam-search integration with the convex head. To address inference details with greater clarity, we have updated the Revision as follows:
> > >
> > > - Appendix G now includes a point-by-point discussion of these three viable inference extensions.
> > > - Appendix B provides an additional (Phase III: Inference) step-by-step walkthrough section of the current decoding procedure.
> > > - We have extended our discussion in Section 7 to specifically highlight decoding-time optimization as a promising direction for future work, and added Key References (below) that are relevant to our inference strategies.
> > >
> > > We hope that these revisions and additional experiments satisfactorily address your questions/concerns. Thank you for taking the time and effort to review our work!
> > >
> > > **Key References:**
> > >
> > > [5] Holtzman, Ari, et al. "The curious case of neural text degeneration." In International Conference on Learning Representations (ICLR), 2020.
> > >
> > > [6] Berdoz, Frédéric, et al. "Alignment-Aware Decoding." arXiv preprint arXiv:2509.26169 (2025).
> > >
> > > [7] Liu, Tianlin, et al. "Decoding-time realignment of language models." In Proceedings of the 41st International Conference on Machine Learning (ICML), 2024.
> > >
> > > [8] Zhu, Jiace, et al. "Path-consistency with prefix enhancement for efficient inference in llms." arXiv preprint arXiv:2409.01281 (2024).

---

### Decision · Program_Chairs · 2026-04-30

**Decision:**

Accept (regular)

**Comment:**

This paper proposes a simple low-cost solution that works. The weaknesses were adequately addressed, thus recommending accept.